# A new adenine nucleotide transporter located in the ER is essential for maintaining the growth of *Toxoplasma gondii*

Senyang Li[1], Jiahui Qian[1], Ming Xu[2], Jing Yang[1], Zhengming He[1], Tongjie Zhao[1], Junlong Zhao[1], Rui Fang[1]*

1 State Key Laboratory of Agricultural Microbiology, College of Veterinary Medicine, Huazhong Agricultural University, Wuhan, Hubei Province, China, 2 Department of Clinical Veterinary Medicine, College of Veterinary Medicine, Huazhong Agricultural University, Wuhan, Hubei Province, China

* fangrui19810705@163.com

**Data Availability Statement:** All relevant data are within the manuscript and its Supporting Information files.

## Abstract

The lumen of the endoplasmic reticulum (ER) is the subcellular site where secretory protein folding, glycosylation and sulfation of membrane-bound proteins, proteoglycans, and lipids occur. The protein folding and degradation in the lumen of the ER require high levels of energy in the form of ATP. Biochemical and genetic approaches show that ATP must first be translocated across ER membrane by particular transporters before serving as substrates and energy sources in the lumenal reactions. Here we describe an ATP/ADP transporter residing in the ER membranes of *T.gondii*. Immunofluorescence (IFA) assay in transgenic TgANT1-HA tag revealed that TgANT1 is a protein specifically expressed in the ER. In vitro assays, functional integration of TgANT in the cytoplasmic membrane of intact *E. coli* cells reveals high specificity for an ATP/ADP antiport. The depletion of TgANT leads to fatal growth defects in *T.gondii*, including a significant slowdown in replication, no visible plaque formation, and reduced ability to invade. We also found that the amino acid mutations in two domains of TgANT lead to the complete loss of its function. Since these two domains are conserved in multiple species, they may share the same transport mechanism. Our results indicate that TgANT is the only ATP/ADP transporter in the ER of *T. gondii*, and the lack of ATP in the ER is the cause of the death of *T. gondii*.

## Author summary

The secretory protein of *T. gondii* is essential for its invasion and normal growth in host cells, all the secretory proteins are synthesized in the ER before being destined for these distinct organelles, such as apicoplast, microneme, dense granule and rhoptry. ER ATP is demanded to support secretory protein folding and trafficking, and the level of ER ATP determines which proteins are able to be directed to the distinct organelles. In theory, the supply of ATP in the ER is necessary for *T. gondii*. However, the transport mechanism and importance of the ER ATP in *T. gondii* are still unclear. In our study, we identified an ATP/ADP transporter (TgANT) located in the ER and verified its function through

**Funding:** The author(s) received no specific funding for this work.

various methods. Unlike the ER ATP/ADP transporter in mammals, we proved that TgANT is functionally specific; the deletion of TgANT caused the interruption of the supply of ATP in the ER, which leads to fatal phenotypic defects of *T. gondii*. Our research further expands the understanding of the growth regulation in *T. gondii*.

## Introduction

The endoplasmic reticulum (ER) is an organelle with complex functions and morphology, mainly responsible for many vital processes such as protein folding, maturation, and degradation. In addition, the ER plays a crucial role in lipid biosynthesis, dynamic $Ca^{2+}$ storage, and detoxification [1]. Many of its critical functions occurring in the ER require an adequate energy supply in the form of adenosine triphosphate, the primary energy source for most cells. In addition, ATP also plays a role in ER as a messenger or auxillary factor to initiate or maintain these functions. Several steps of protein folding in the ER, such as the formation of disulfide bonds, require a continuous supply of ATP to meet its energy requirements [2,3]. It has been demonstrated that ATP binding and hydrolysis are required to drive the activity of ER chaperons, and protein processing in the ER is suspended under the condition of ATP depletion. Besides, protein misfolding is proved to be the primary cause of many human diseases, including Parkinson's, Alzheimer's, and other age-related diseases [4–6].

In eukaryotes, Oxidative phosphorylation is a predominant and highly efficient metabolic strategy producing large amounts of ATP, which is exchanged with ADP through mitochondrial ADP/ATP carrier (AAC) across the inner mitochondrial membrane [7]. Under hypoxic conditions, most eukaryotes can utilize the glycolysis process, which takes place in the cytoplasm, to maintain ATP levels. Studies have shown that many types of tumor cells have a more vital ability to grow and proliferate by enhancing the glycolysis process [8]. Furthermore, both the size and charge of the nucleotides prevent free flow through the bio-membrane, which makes cytoplasmic ATP into the ER lumen must pass through a specific transport protein and transfer ADP out of the ER. Although the functional principle of the ER-resident protein and the critical ER-related processes had been clarified, the mechanism of ER ATP transport is still a mystery.

The current research on the ER ATP/ADP transporter is still in its infancy. In yeast, the ATP transporter in the ER was identified by reconstituted into proteoliposomes, a 68-kDa protein (Sac1p) was believed to play a vital role for ATP transport during protein translocation [9]. However, it turns out that Sac1p itself is not an ATP transporter because the purified Sac1p reconstituted into proteoliposomes does not catalyze any ATP uptake. It may act as an essential regulator of the transport process by controlling the ER inosine phosphate [10]. The lack of Sac1p results in the defect of microsomal ATP transport, but the microsomes of the sac1Δ strain still retain a 15% ATP transport rate, indicating the existence of another independent ATP transporter in the microsomal membrane [9,11]. In plants, only one document describes the ATP transporter ER-ANT1 in *Arabidopsis thaliana*. Functional integration of ER-ANT1 in the cytoplasmic membrane of intact *E. coli* cells reveals high specificity for an ATP/ADP antiport. Disruption of ER-ANT1 results in a catastrophic phenotype of the plant, which is characterized by significant growth retardation, impaired root and seed development [12]. Interestingly, all these effects observed with the ER-ANT1 mutants can be reverted to a wild-type-like phenotype by plant growth at a high external $CO_2$ concentration [13]. The mammalian ER ATP/ADP transporter remained elusive until a recent article determined that SLC35B1 (AXER) is the recognized mammalian ER ATP/ADP transporter. Unlike the severe

phenotypic defects of *Arabidopsis thaliana* caused by ER-ANT1, the depletion of AXER did not cause severe growth damage to the HeLa cells, it only caused the activity of the BIP protein to decrease and slight cell growth slowed down. Whether this indicates the existence of other transport mechanisms in the ER of mammalian cells remains to be further verified [14].

There are no reports on the ER ATP transporter in the apicomplexan parasite and other parasite species. It has been revealed that an SLC35B1 family protein (HUT1), which is homologous to AXER, involves in larval developments in *Caenorhabditis elegans*. The hut-1 deletion mutant and RNAi worms showed larval growth defect and lethality with disrupted intestinal morphology. However, lethal phenotype defects and the ER stress of the mutant can be rescued with the human hut-1 ortholog UGTrel1 (AXER). Except for AXER, heterologous supplementation of a variety of nucleotide sugars cannot compensate for the phenotypic defect caused by HUT-1, which suggests that HUT-1 may be an ER ATP/ADP transporter in *Caenorhabditis elegans*, but further experimental verification is needed [15].

*Toxoplasma gondii* is an obligate intracellular parasitic protozoan that infects more than two billion people worldwide. In humans, toxoplasmosis can cause serious complications, particularly in fetuses and immunocompromised patients. In the process of host-pathogen interactions, *T.gondii* utilizes a large amount of specialized secreted proteins to modify host cytokines to promote invasion and replication, such as microneme protein (MICs), rhoptry protein (ROPs), and dense granules protein (GRAs) [16–19]. Therefore, the continuous and stable supply of ATP from the ER is essential for the process and synthesis of secreted proteins. However, how the ER of *T.gondii* maintains a stable ATP supply is unknown. Here we identified an ATP/ADP transporter TgANT located in the ER of *T.gondii*. For functional characterization of TgANT, we expression TgANT in *E. coli*, this system has been shown to functionally integrate several membrane proteins into the bacterial cytoplasmic membrane [12,14,20]. Functional integration of TgANT in the cytoplasmic membrane of intact *E. coli* cells reveals high specificity for an ATP/ADP antiport. In addition, our results also indicate that the severe growth defects of *T. gondii* were caused by the lack of ATP in the ER.

## Results

### TgANT is an ER membrane protein in *T. gondii*

Due to the lack of research on the ER ATP/ADP transporter, there is no accurate information for gene screening in *T.gondii*. Therefore, the BLAST algorithms were used for searching the *T. gondii* ER ATP/ADP transporter with the ER-ANT1 and AXER in *ToxoDB* (https://toxodb.org/toxo/app/). Blastp analysis revealed only three sequences encoding putative solute carriers family protein (*TGGT1_273390*, *TGGT1_300360*, *TGGT1_249900*) were high homologous to ER-ANT1, but no genes with homology to AXER were found. Since ER-ANT1 has high homology (more than 60%) with the mitochondrial ATP/ADP transporter of Arabidopsis [12], these three proteins in *T. gondii* found by Blastp analysis are also located in mitochondria, rather than located in the ER (S1A Fig). Meanwhile, we screened sixty-nine sequences (S1 Table) by setting the CRISPR phenotype value and transmembrane domain (TMD) in the *ToxoDB* database. Among the sixty-nine selected genes, we focused on *TGGT1_254480*, which was annotated as UDP-galactose transporter family protein in the *ToxoDB* database, and predicted to have ten transmembrane helices (Fig 1A and 1B). Besides, *TGGT1_254480* contains a C-terminal dilysine motif (-KKQC) that serves as an ER retention signal, C-terminally dilysine motif shown in other proteins to limit exit from the ER (Fig 1A) [21]. According to the nomenclature in other species, we named *TGGT1_254480* as *T. gondii* Adenine Nucleotide Transporter (TgANT), and distinguished from mitochondrial ATP/ADP carrier (AACs).

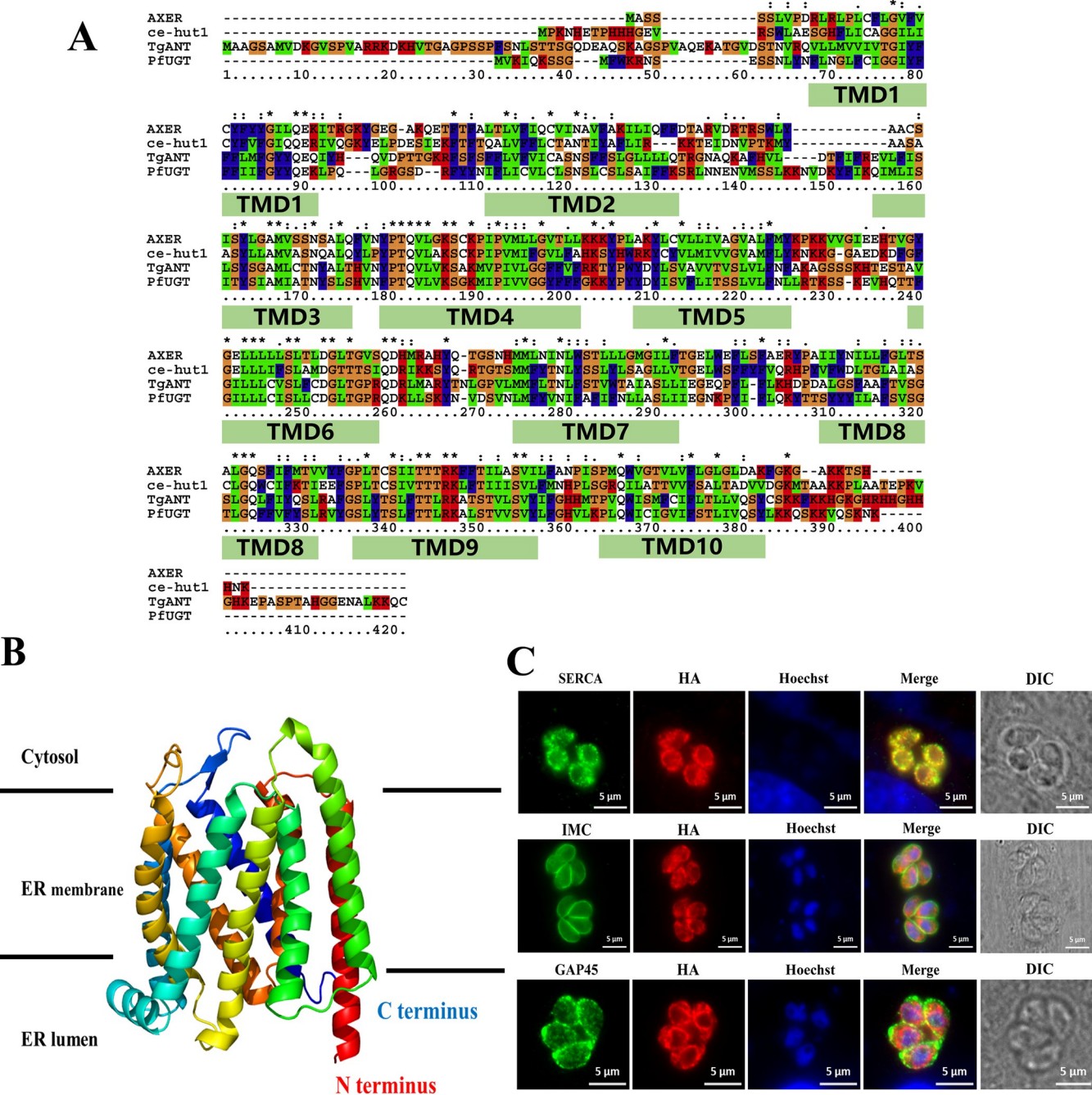

**Fig 1. Putative structure and intracellular localization of TgANT. (A)** Protein sequences were from NCBI and shown in Gene ID code for Homo sapiens (AXER, Gene ID: 10237), *Caenorhabditis elegans* (hut-1, Gene ID: 176690), plasmodium (PF3D7-1113300, Gene ID: 810688). The sequences were aligned using ClustalX and TMHMM servers. **(B)** The hypothetical structural model of TgANT was predicted by the Phyre2 server. The nitrogen terminus and carboxy terminus face the ER lumen, the double lysine motif (-KKQC) is located near the C-terminal end of TgANT. **(C)** TgANT-HA parasites were subjected to immunostaining using antibodies against the HA tag along with SERCA (sarco/endoplasmic reticulum Ca²⁺-ATPase, ER protein), The result shows that TgANT was complete co-localization with the ER-resident protein SERCA. But not co-localization with IMC (inner membrane complex, inner membrane protein) and GAP45 (gliding-associated protein 45, plasma membrane protein) of *T. gondii*. Scale bars: 5 μm.

We first tried to prepare TgANT polyclonal antibodies using the three peptide fragments outside the transmembrane region of TgANT exposed to the cytoplasm and ER lumen. However, the polyclonal antibody we prepared could not detect TgANT protein, either by Western

blot or IFA. To determine the subcellular location of TgANT in *T.gondii*, 10×HA-tagged was cloned into the C-terminus by CRISPR-Cas9–mediated site-specific integration in the RHΔku80 strain (TgANT-10×HA) and diagnostic PCRs confirmed the correct integration of the 10×HA-tagged. IFA staining showed that TgANT was located in the ER and complete co-localization with the ER-resident protein SERCA (S1B Fig). Since TgANT has a C-terminal polylysine motif (-KKQC), this sequence was considered to be potentially responsible for ER retention [21]. Therefore, to avoid the addition of the C-terminal HA epitope affecting the correct localization of TgANT, we used CRISPR/cas9 technology to insert the HA epitope outside the transmembrane region of TgANT (AGSSSK-HA-HTESTA, TgANT-HA) and expressed it at the *UPRT* (uracil phosphoribosyl-transferase) site of TATi strain. IFA showed that TgANT was also complete co-localized with the ER-resident protein SERCA (Fig 1C).

## Functional expression of TgANT in *E. coli* cells

Previous studies have shown that the heterologous synthesis of mitochondrial and ER ATP/ADP transporter in *E. coli* results in their functional integration into the bacterial cytoplasmic membrane. Its transport characteristics are similar to those obtained from the authentic membrane [12,14,20]. To test whether TgANT might act as an ER ATP/ADP transporter, we cloned the cDNA of TgANT into the plasmid pet16b and expressed it in *E. coli* (BL21, DE3). After iso-propyl-β-D-thiogalactopyramoside (IPTG) induction, we investigated whether the TgANT is integrated into the plasma membrane using IFA. As expected, the TgANT was expressed and inserted into the bacterial membrane (Fig 2A). Uptake studies were carried out with radioactively labeled [α-$^{32}$P] ATP. The result showed ATP was imported into intact bacterial cells harboring TgANT and revealed a very high transport capacity for radioactive ATP. By contrast, the uninduced *E. coli* was not able to absorb ATP at significant rates (Fig 2B). Competition experiments with different potential substrates showed that the heterologously expressed TgANT in *E. coli* were highly specific for ATP and ADP, with no competition from AMP, CTP, GTP, UTP, UDP-glucose, or UDP-galactose (putative substrate) with [α-$^{32}$P] ATP import. The ability of *E. coli* to take up radioactive ATP was inhibited in the presence of non-radioactive ATP or ADP, which further proves that TgANT can import ATP as well as ADP (Table 1).

## TgANT is an ATP/ADP antiporter

Mitochondrial ADP/ATP Carrier (AACs) work in counter exchange (antiport mode), allowing ADP and ATP to flow in or out at a stoichiometric ratio of 1:1. To investigate the antiport mode of TgANT, we carried out back exchange experiments with *E. coli* cells preloaded with radioactively labeled [α-$^{32}$P] ATP. If TgANT was maintained an antiport mode in induced *E. coli* cells, the newly imported radioactively labeled [α-$^{32}$P] ATP could be driven out of the cells by unlabeled ATP and ADP. Due to metabolic activity, the [α-$^{32}$P] ATP of *E. coli* cells is converted into labeled [α-$^{32}$P] ADP.

First, the radioactively labeled nucleotides, which were chased from the cells by the addition of an excess of unlabeled ATP and ADP, were analyzed by thin-layer chromatography (TLC). After *E. coli* was preloaded with radioactive [α-$^{32}$P] ATP, we performed a back exchange experiment using unlabeled ATP and ADP. The results showed that no significant radioactivity was detected after incubation in Phosphate Buffer solution (PBS) (Fig 3A, lane 1). In the presence of nonlabeled ATP or ADP, radioactively labeled adenine nucleotides were driven out by TgANT (Fig 3A, lane 2 ADP, lane 3 ATP). Both radioactively labeled [α-$^{32}$P] ATP and [α-$^{32}$P] ADP were detected from the supernatant of disrupted *E. coli* (Fig 3A, lane 4). In addition, after *E. coli* absorbs radioactively labeled [α-$^{32}$P]-ATP for 15 minutes, additional excess unlabeled ATP or ADP was used to chase the radioactive ATP. Then, the remaining radioactivity in *E. coli* was

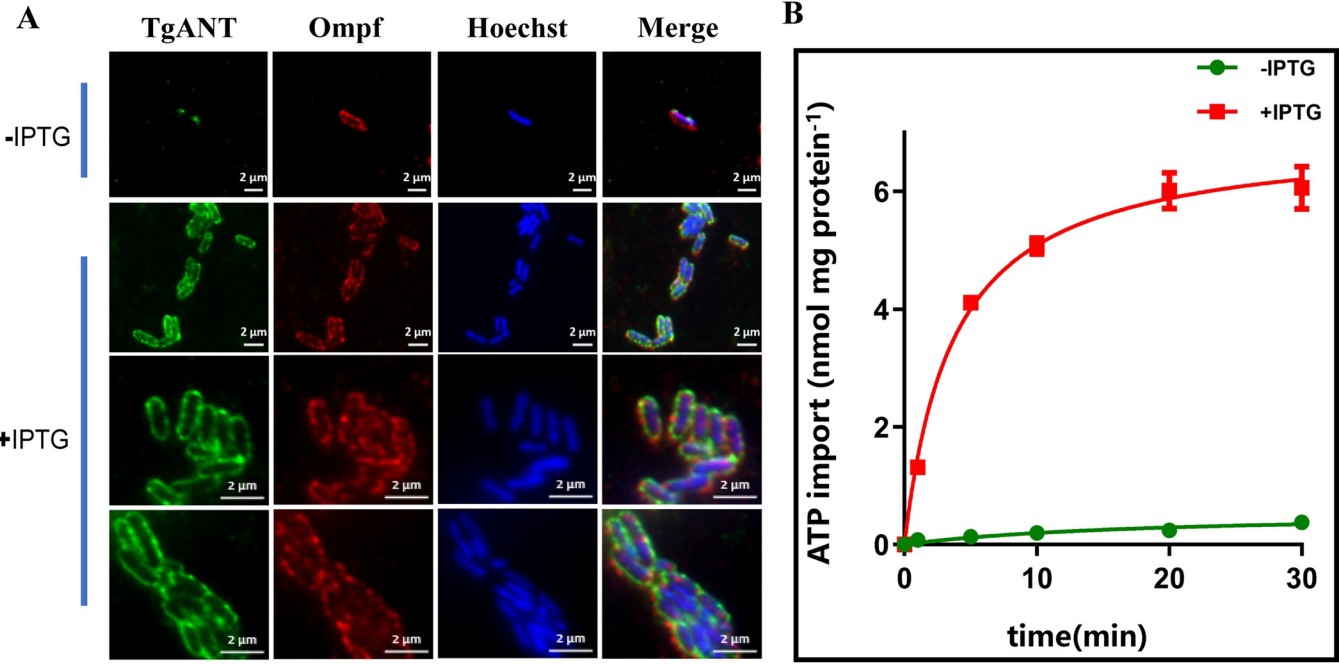

**Fig 2. Heterologously expressed TgANT is an ATP and ADP carrier. (A)** The *E. coil* harboring pet16b-TgANT-HA were subjected IFA using antibodies against the HA tag, Ompf (outer membrane porin F) and Hoechst. The result shows that TgANT was expressed in the membrane of *E. coil*. Scale bars: 2 μm. **(B)** Time course for [α-$^{32}$P] ATP uptake into intact *E. coli* cells. IPTG-induced *E. coli* cells harboring the plasmid encoding TgANT were incubated with 25 nM [α-$^{32}$P] ATP for the indicated time intervals. Noninduced *E. coli* cells with the plasmid encoding TgANT were used as control. Means ± s.e.m., three independent assays.

detected at a specific time point. The results show that the additional unlabeled ATP (Fig 3B) and ADP (Fig 3C) both can cause the rapid outflow of labeled nucleotides in *E. coli*. These results support the truth that TgANT acts as antiporters like Mitochondrial AACs.

## TgANT is important for parasite growth

Since TgANT has a low phenotype score (-4.8), it is very important for *T.gondii* and cannot be deleted directly from the parasite [16]. Therefore, an inducible knockdown system (iTgANT)

**Table 1. Impact of potential substrates and effectors on ATP transport by TgANT.**

| Inhibitor | Uptake(% of control) | ± SEM (%) |
|---|---|---|
| ADP | **3.9** | **±0.31** |
| ATP | **3.4** | **±0.25** |
| UDP-Gal | 91.4 | ±4.38 |
| UDP-Glu | 99.2 | ±2.04 |
| UTP | 95.8 | ±3.47 |
| AMP | 107.7 | ±1.71 |
| CTP | 105 | ±4.95 |
| GTP | 90.4 | ±6.25 |

To test the substrate specificity of TgANT, a mixture of 25 nM [α-32P] ATP and non-radioactive potential substrate at one-hundred-fold concentration was used to detect the inhibitory effect. *E. coli* cells expressing TgANT uptake of 25 nM of [α-$^{32}$P] ATP for 15 min were used as control. Means ± s.e.m., three independent assays.

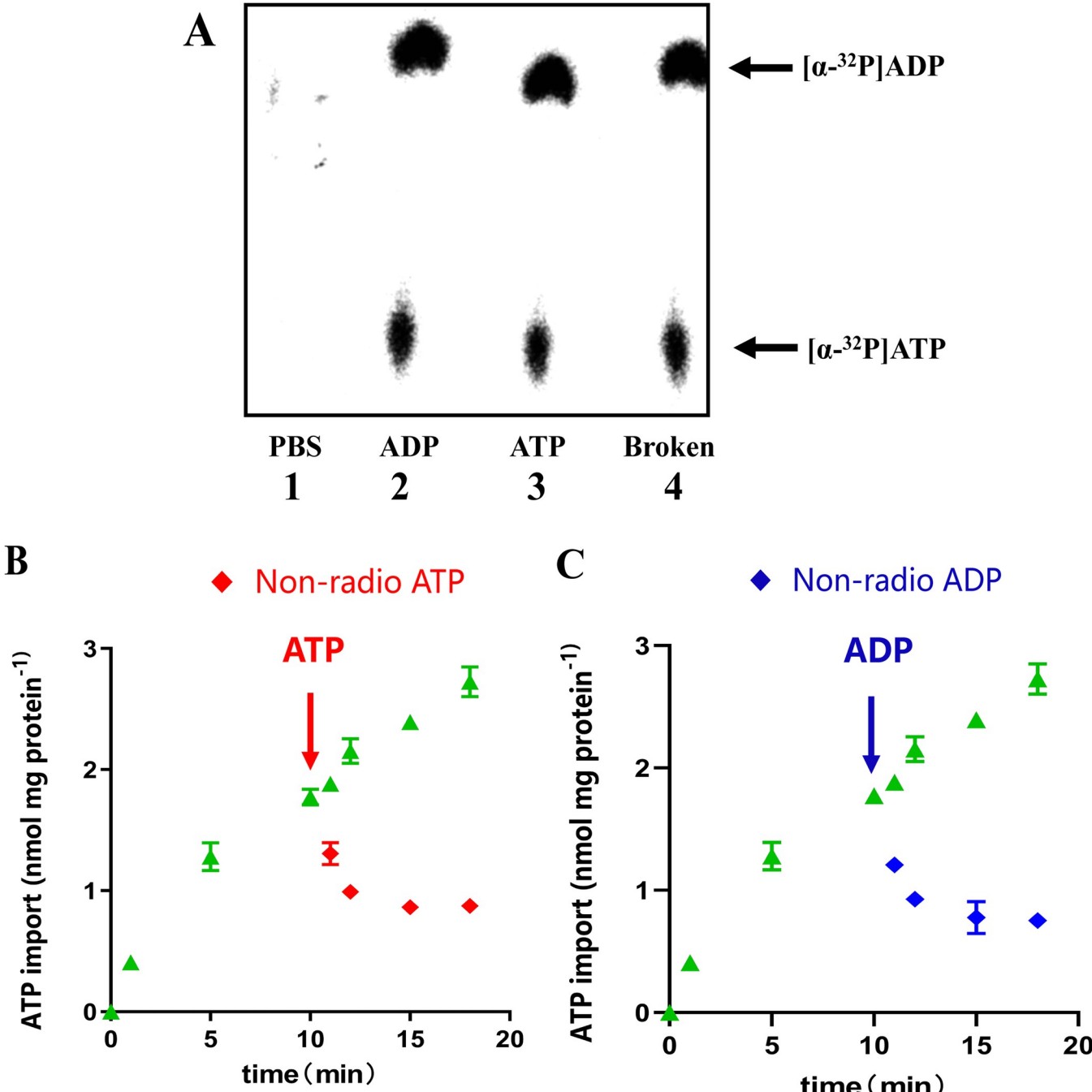

**Fig 3. Back exchange experiments with *E. coli* cells. (A)** *E. coli* cells expressing TgANT were preloaded with 50nM radioactive [α-³²P] ATP for 5 min, followed by carried out back exchange experiment with 5 μM non-labeled ATP, ADP, or PBS for 3 min and terminated by rapid centrifugation. Then, 2 μL sample of the supernatant were loaded onto the TLC PEI cellulose plate. The supernatant of the broken *E. coli* cells was used to detect the metabolic activity (lane 4). **(B)** and **(C)** *E. coli* cells harboring the plasmid encoding TgANT were incubated with 50 nM [α-³²P] ATP for 10 min. Then, efflux was induced by the addition of fifty-fold unlabeled ATP **(B)** or ADP **(C)** for 1min, 2min, 5min and 8min. Means ± s.e.m., three independent assays.

was used to deplete TgANT expression, where the TgANT native promoter was replaced with an anhydrotetracycline (ATc)-dependent promoter (Fig 4A). A Ty epitope was cloned in the N-terminal of TgANT to detect the depletion of TgANT [22]. Specific primers across the homology arms were used to confirm the mutant strains' promoter integration (Fig 4B).

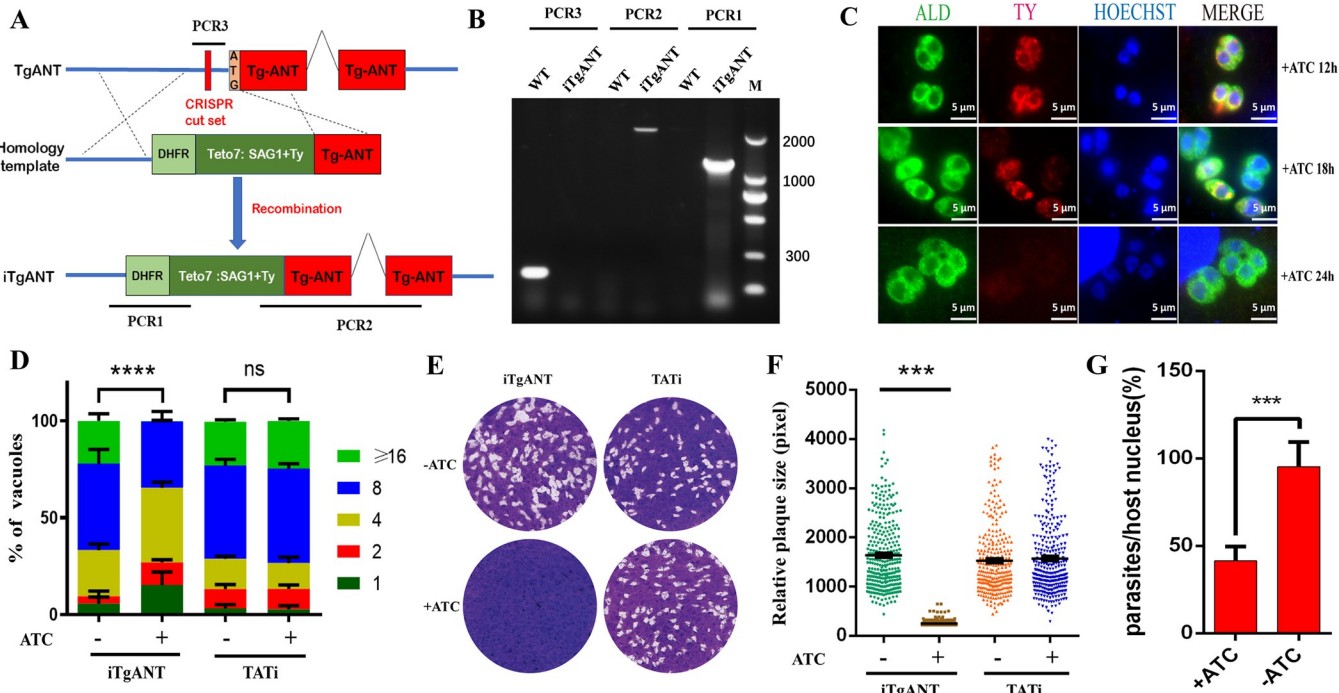

**Fig 4. Conditional knockdown of TgANT results in severe growth defects. (A) and (B)** schematic showing generation of the iTgANT strains via CRISPR-Cas9–assisted gene engineering, which was done by replacing the endogenous TgANT promoter with a tetracycline regulatable promoter (SAG1:: TetO7) in the TATi line **(A)**. PCR1–PCR3 indicates screening of clonal mutant **(B)**. **(C)** The Ty epitope at the N-terminal of TgANT was detected by IFA (red) at the appointed time after the 0.5 μg/ml ATc treatment. The results showed that TgANT was completely depletion after 24 hours of drug treatment. The fructose-1,6-bisphosphate aldolase (ALD) of *T. gondii* as a marker. **(D)** Intracellular replication assay comparing parasite proliferation under indicated conditions. TATi and iTgANT strains were pretreated with 0.5 μg/ml ATc for 40 h or left untreated. Subsequently they were allowed to infect HFF monolayers for 40–60 min, and uninvaded parasites were washed with PBS. Whereafter, invaded parasites were cultured with 0.5 μg/ml ATc or left untreated for 24 h to determine the number of parasites in each parasitophorous vacuole (PV). ***p ≤ 0.001, two-way ANOVA, three independent repeated, a representative one is shown here. **(E) and (F)** Parasites were grown ±0.5 μg/ml ATc for 7 days to form plaques on HFF monolayers **(E)**. plaque size presented as relative area (pixel size calculated by Photoshop) from three independent assays **(F)**. ***p ≤ 0.001, Student's t-test, three independent assays. **(G)** Host cell invasion efficiencies determined by a two-color staining protocol that distinguishes intracellular from extracellular parasites. **p ≤ 0.01, Student's t-test, three independent repeated, a representative one is shown here.

Because TgANT has multiple transmembrane domains and both the N-terminal and C-terminal face the ER lumen, it was failed to detect Ty epitope by western blot. Therefore, the expression of the TgANT after ATc treatment was monitored by IFA using the specific antibody of Ty epitope, which revealed that the TgANT was almost undetectable after 24h treatment with 0.5 μg/ml ATc (Fig 4C). Next, iTgANT growth phenotype was assessed by replication assay (Fig 4D) and plaque assay (Fig 4E and 4F). For the plaque assay, the iTgANT strains treated with 0.5 μg/ml ATc did not form any visible plaques compared with the iTgANT strains without ATc treatment. Moreover, after treatment with ATc, the intracellular replication speed of iTgANT strains was also significantly reduced. Taken together, these data indicate that TgANT is essential for *Toxoplasma's* normal growth.

Because all the secretory proteins are synthesized in the ER before being destined for these distinct organelles [23,24] and ER ATP is essential to support protein chaperone functions for protein folding and trafficking, the level of ER ATP determines which proteins are able to transit to the cell surface [25–27]. Besides, the secreted protein of *T gondii* is essential for its invasion and normal growth in host cells. Therefore, we speculate that the lack of TgANT not only severely affects the intracellular replication of *T.gondii* but also affects the ability to invade host cells. So, we next examined the ability to invade host cells in the iTgANT mutant. As expected,

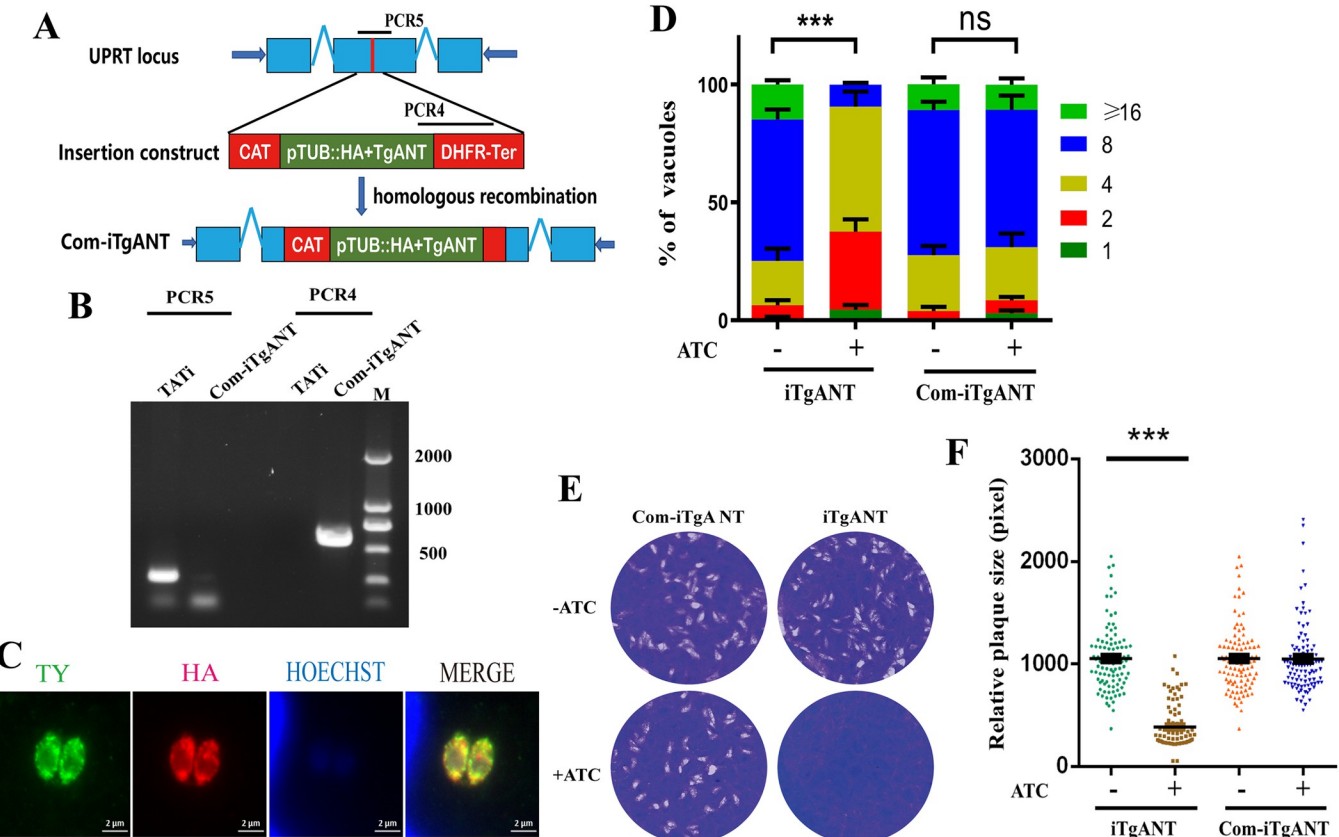

**Fig 5. TgANT complementation fully restored the growth defects of TgANT depletion mutants. (A) and (B)** schematic showing generation of the iTgANT strains via CRISPR-Cas9–assisted gene engineering**(A)**. PCR1–PCR3 indicates screening of clonal mutant **(B)**. **(C)** IFA confirmed that the ectopic TgANT (red) was successfully expressed and localized to the ER. **(D)** Intracellular replication assay comparing parasite proliferation under indicated conditions. TATi and Com-iTgANT strains were pretreated with 0.5 μg/ml ATc for 40 h or left untreated. Subsequently, they were allowed to infect HFF monolayers for 40–60 min, and uninvaded parasites were washed with PBS. Whereafter, invaded parasites were cultured with 0.5 μg/ml ATc or left untreated for 24 h to determine the number of parasites in each parasitophorous vacuole (PV). ***p ≤ 0.001, two-way ANOVA, three independent repeated, a representative one is shown here. **(E) and (F)** Plaque assays comparing the growth of TgANT depletion strain and TgANT complementation strain. Parasites were grown ±0.5 μg/ml ATc for 7 days to form plaques on HFF monolayers **(E)**. plaque size presented as relative area (pixel size calculated by Photoshop) from three independent assays **(F)** ***p ≤ 0.001, Student's t-test, three independent repeated.

compared with the iTgANT mutant strain without ATc treatment, the ability to invade host cells of the iTgANT mutant strain was significantly reduced after 36h of ATc treatment (Fig 4G).

To determine the observed phenotypic specificity in the iTgANT mutant strain, we complemented the iTgANT mutant strain with N-terminally HA-tagged TgANT expressed from the *UPRT* locus (Fig 5A) [28]. The complemented strain (Com-iTgANT) was confirmed by screening PCRs and IFA, diagnostic PCRs confirmed the desired integration of the TgANT in *UPRT* locus (Fig 5B) and IFA confirmed that the ectopic TgANT was successfully expressed and localized to the ER, just like the endogenous protein (Fig 5C). As expected, ectopic expression of TgANT completely rescued the phenotypic defects observed in the iTgANT mutant strain, confirming that TgANT is indeed required for normal growth of the parasites (Fig 5D, 5E and 5F).

## TgANT is not a UDP-galactose transporter

Just as AXER was previously annotated as UDP-galactose transporter in GenBank, TgANT is also annotated as UDP-galactose transporter (UGT) family protein in the *ToxoDB* database.

Before that, UDP-galactose was considered to be their potential substrate. Lectin experiments were performed to further prove that TgANT does not have the function of UGT and also verify the experimental results in *E. coli* cells.

Alexa Fluor 488 conjugate lectin GS-II (GS-II-488) was used to detect whether TgANT could transport UDP galactose. Lectin GS-II contained a single binding site specific for terminal non-reducing α-or β-linked *N*-acetyl-D-glucosamine (GlcNAc). In wild-type CHO cells, galactose was normally attached to GlcNAc residues, whereas the mutant CHO cells (Lec8) were deficient in UGT so that GlcNAc residues were more exposed on the cell surface of UGT-deficient Lec8 cells due to decreased galactosylation [29]. It has been shown by lectin staining that Lec8 cells are significantly defective in the transport of UDP-galactose and exhibit a 70–90% deficiency of galactose attached to glycoproteins, glycolipids, and some proteoglycans [29,30].

We cloned the CDS of TgANT and hUGT2 (positive control) with an N-HA epitope into the pCDNA3.1 eukaryotic expression vector and then transiently expressed in Lec8 cells. Following, the cells were examined under a microscope to determine whether they were bounded with GS-II-488. IFA results showed that no green fluorescence was detected in Lec8 cells which successfully expressed hUGT2, indicating that hUGT2 can complement the galactosylation defect of Lec8 cells. On the contrary, intensely green fluorescence can still be detected in lec8 cells which successfully express TgANT (S1C Fig). The results indicate that hUGT2 but not TgANT has the function of UDP-galactose transporter. We also used lectin blot to detect the level of galactosylation in *T. gondii* after the TgANT gene was depleted. The results showed that the level of galactosylation in the tachyzoites did not change after the depletion of TgANT (S1D Fig).

Besides, we attempted to complement the iTgANT mutant strain with N-terminal HA-tagged hUGT2 expressed from the *UPRT* locus. The successful construct of complemented strain (iTgANT::hUGT2) was confirmed by screening PCRs, and IFA demonstrate that hUGT2 was successfully expressed in ER (S2A Fig). Next, iTgANT::hUGT2 strains growth phenotype was assessed by plaque assay and replication assay. However, it turned out that the expression of hUGT2 failed to save the phenotypic defect of the iTgANT strain (S2B and S2C Fig). These results further indicate that TgANT is an ATP/ADP transporter in the ER of *T.gondii*, rather than a UDP-galactose transporter.

## The lack of ATP in the ER maybe caused the death of *T. gondii*

As a protozoan with specialized secretion ability, *T.gondii* utilizes a large amount of specialized secreted proteins to modify host cytokines to promote invasion and replication. We also wanted to figure out whether the growth defect of *T. gondii* was caused by the depletion of ATP in the ER, or the depletion of TgANT caused other unknown changes. Therefore, we attempted to confirm the ATP and ADP binding sites of the TgANT through the mitochondrial ADP/ATP Carrier sequences of *Arabidopsis thaliana*, *Human*, and *Saccharomyces cerevisiae*.

Mitochondrial ADP/ATP Carriers remarkably conserved in animals and plants consist of three homologous sequence repeats of about 100 amino acids [31], which form a 3-fold pseudosymmetrical structure with the translocation path through the center of the molecule [32]. However, it is different from *Arabidopsis thaliana* ER-ANT1, which has high homology with mitochondria ATP/ADP transporters. We found that both TgANT and AXER had almost no homology to these mitochondrial ADP/ATP Carriers. Accordingly, it was suspected that there might be different recognition and transport mechanisms between the ER ATP/ADP and the mitochondrial ATP/ADP transporters. The BLAST algorithm was used to search the homology sequences of TgANT in NCBI to find the conserved protein domain. We consider that

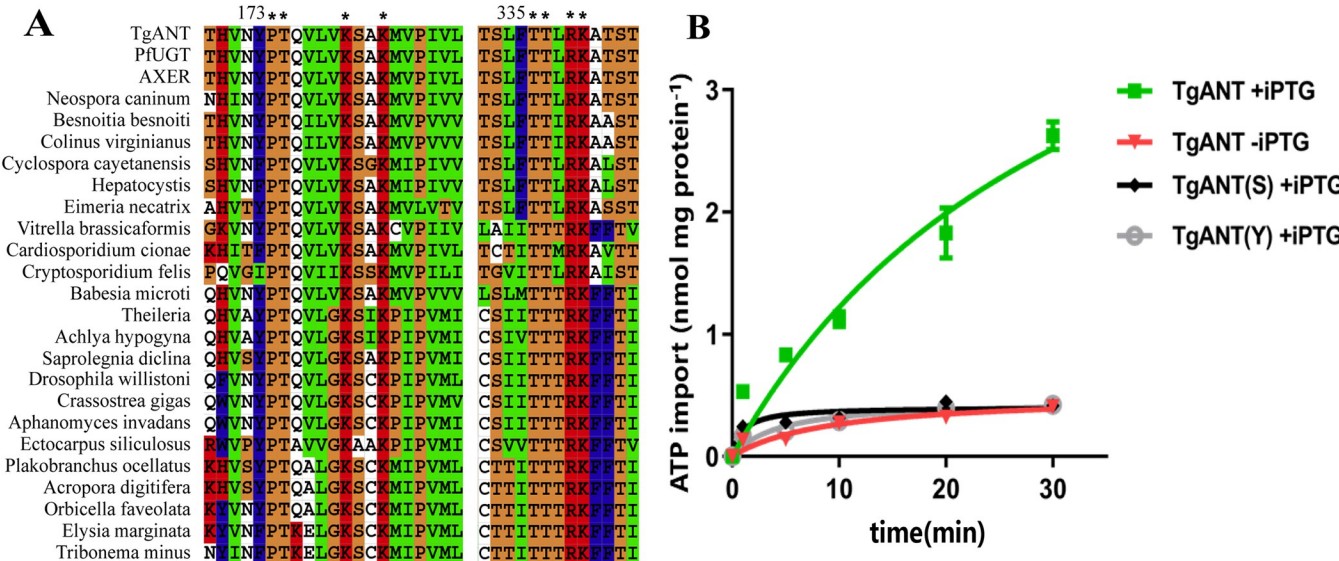

**Fig 6. Conserved protein domain prediction and the absorb experiment of *E. coli* cells harboring mutant plasmid. (A)** Conserved protein domain of TgANT were searched in NCBI. These two conserved domains are conserved in many species, such as mollusks, fungal, drosophila willistoni. **(B)** IPTG-induced *E. coli* cells harboring the plasmid encoding mutant cDNA of TgANT (pet16b-TgANT-[173]YSxxxxQxxQ, Gray hollow dots, and pet16b-TgANT-[335]SSxQQ, Black dot) were incubated with 25 nM [$\alpha$-[32]P] ATP for the indicated time intervals. IPTG-induced or Noninduced *E. coli* cells harboring the plasmid encoding wild-type TgANT were used as control. Means ± s.e.m., data of three independent repeated.

these conserved protein domains play a crucial role in the ATP/ADP transport of TgANT in ER. By homology analysis, two highly conserved protein domains [173]PTxxxxKxxk and [335]TTxRK were found in TgANT (Fig 6A).

To avoid the destruction of three-dimensioanl structure of TgANT, the conserved amino acids in the two highly conserved protein domains were mutated using amino acids with similar hydrophilic or hydrophobic properties, and then the mutant cDNA of TgANT was cloned into the pet16b plasmid (pet16b-TgANT-[173]YSxxxxQxxQ and pet16b-TgANT-[335]SSxQQ) and expressed in *E. coli*. Uptake experiments were carried out with radioactively labeled [$\alpha$-[32]P] ATP as before. As expected, the two *E. coli* expressing mutant plasmids no longer had the ability to import ATP (Fig 6B), which indicates that these two key domains are essential for TgANT to transport ATP and ADP.

It was already confirmed that the mutation of the conserved domains made TgANT expressed in *E.coli* unable to transport ATP. However, it was still unknown whether the mutate TgANT could keep the parasite with phenotypic defect (iTgANT strain) alive without the function of ATP transport. In another word, if the iTgANT strain, which is lack TgANT, can survive after being complemented with the mutant TgANT, it means that the ATP transported by TgANT is not necessary for *T. gondii*, and TgANT may have other unknown functions to maintain the growth. Therefore, we complemented the iTgANT strain with the mutant cDNA of TgANT (Comp-TgANT-[173]YSxxxxQxxQ and Comp-TgANT-[335]SSxQQ) in the *UPRT* locus. IFA assay confirmed that the two mutant strains were still localized in the ER (Fig 7A). As expected, mutations in the two conserved protein domains resulted in TgANT completely losing function. Moreover, in result neither of the two mutant strains could rescue the phenotypic defects caused by the depletion of TgANT (Fig 7B and 7C).

AXER is highly specific for ATP and ADP, with no competition from AMP, CTP, GTP, UTP, UDP-glucose, or UDP-galactose (its putative substrate) with [$\alpha$-[32]P] ATP import [14]. To further determine the severe phenotypic defect of the iTgANT strain is caused by the lack

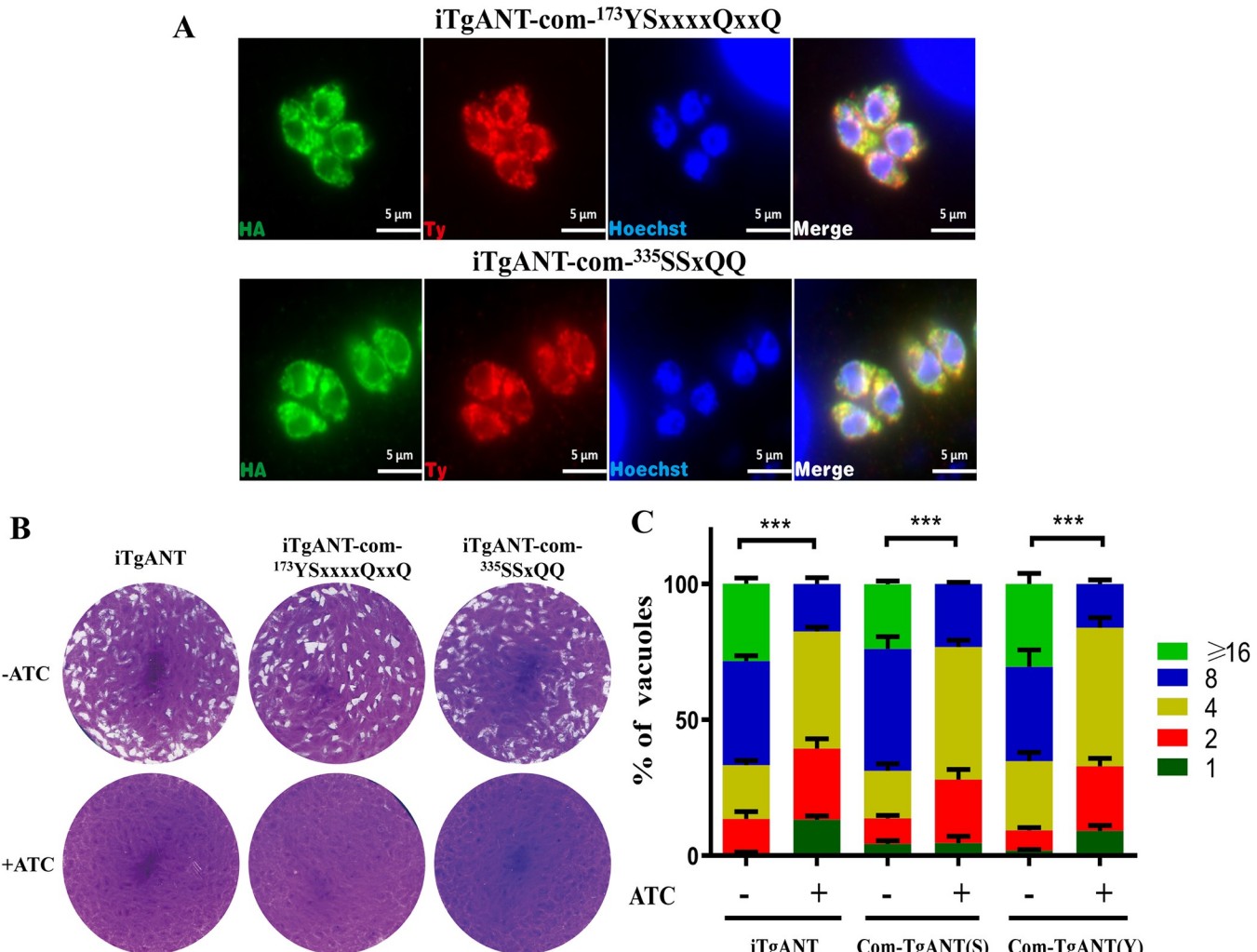

**Fig 7. The phenotypic experiment of TgANT compensation mutant strains. (A)** IFA confirmed the correct integration and expression in ER of Com-TgANT-$^{173}$YSxxxxQxxQ strain and Com-TgANT-$^{335}$SSxQQ strain.HA denote the mutant CDS of TgANT, and Ty denotes the TgANT in situ. **(B)** plaques experiment of iTgANT strains and mutant compensation strains, parasites were grown ±0.5 µg/ml ATc for 7 days to form plaques on HFF monolayers. **(C)** iTgANT strains and mutant compensation strains (Com-TgANT(Y) refer to Com-TgANT-$^{173}$YSxxxxQxxQ strain and Com-TgANT(S) refer to Com-TgANT-$^{335}$SSxQQ) were pretreated with 0.5 µg/ml ATc for 40 h or left untreated. Subsequently they were allowed to infect HFF monolayers for 40–60 min, and uninvaded parasites were washed with PBS. Whereafter, invaded parasites were cultured with 0.5 µg/ml ATc or left untreated for 24 h to determine the number of parasites in each parasitophorous vacuole (PV). ***$p \leq 0.001$, two-way ANOVA, three independent repeated, a representative one is shown here.

of ATP in the ER, heterologous supplementation was performed with AXER. First, the iTgANT mutant strain was complemented with N-terminally HA-tagged AXER expressed in the *UPRT* locus. However, both transcript variants of AXER (SLC35B1 and SLC35B1/Isoform 2) were located in apicoplast but not ER. Therefore, sixty-three amino acids from the C-terminal of TgANT were used as the guide sequence integration with SLC35B1 and then expressed in the *UPRT* locus in iTgANT mutant strain (iTgANT::AXER). Specific primers and IFA confirmed the correct integration and expression in ER (Fig 8A). As expected, ectopic expression of AXER completely rescued the phenotypic defects observed in the iTgANT mutant strain (Fig 8B and 8C).

According to these results, is reasonable to presume that these two conserved domains are very likely to be the substrate-binding sites of TgANT. Additionally, we believe that the

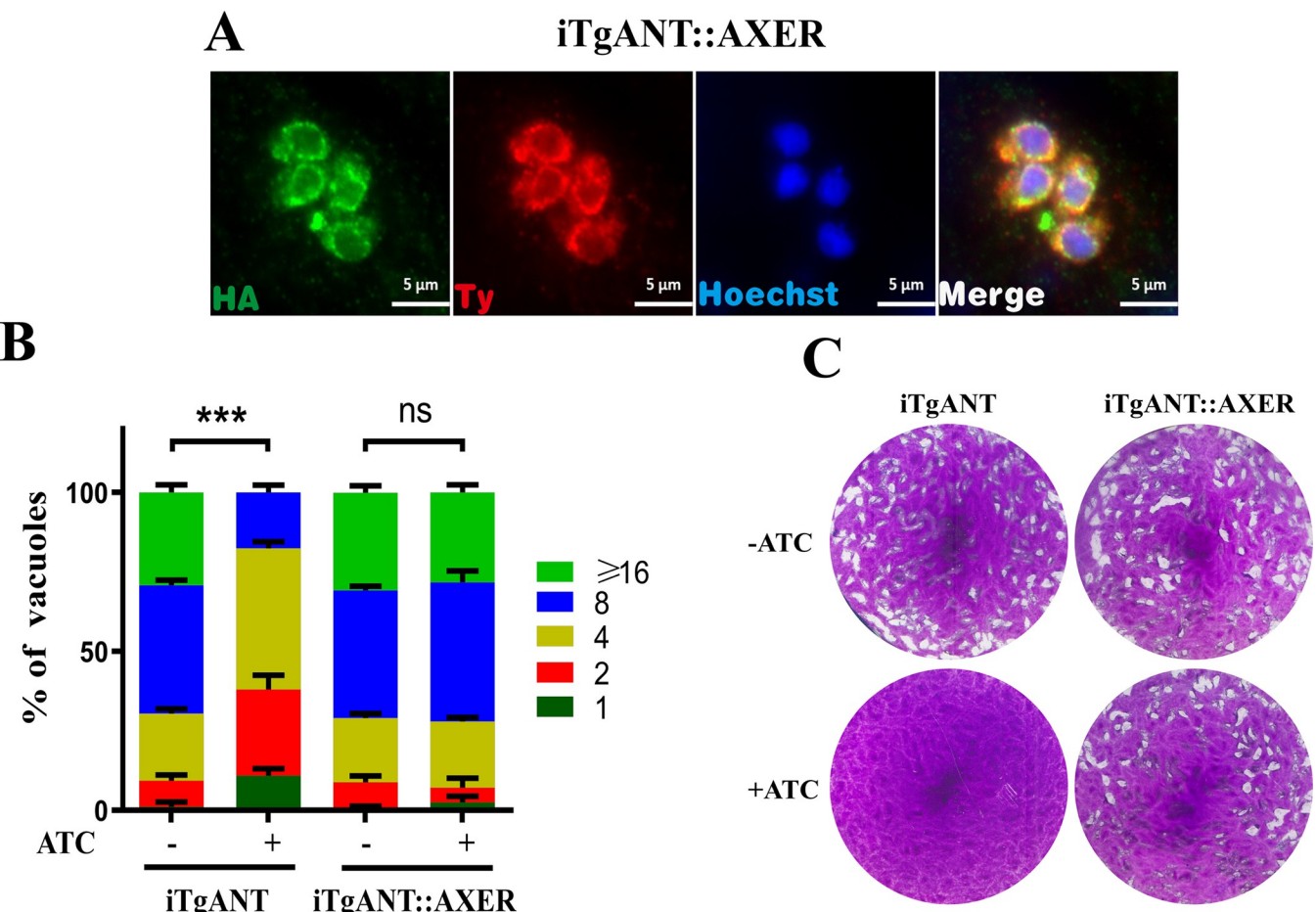

**Fig 8. The Phenotypic experiment of heterologous supplementation strains. (A)** IFA confirmed the correct integration and expression in ER of iTgANT:: AXER strain. **(B) and (C)** Intracellular replication assay **(B)** and plaque assay **(C)** comparing the growth of TgANT depletion strain and AXER complementation strain. ***p ≤ 0.001, two-way ANOVA, three independent repeated, a representative one is shown here.

depletion of TgANT resulted in the severe growth defects of *T.gondii*, due to the the lack of ATP in the ER.

## PfUGT(PfANT) is also a nucleotide transporter in the ER of *P. falciparum*

In *P. falciparum*, PfUGT (*PF3D7_1113300*) is highly homologous to TgANT (Fig 9A), which was also considered a UDP-galactose transporter located in the ER. The same method was used to verify its transport function. As expected, PfUGT also revealed a transport capacity for ATP and ADP (Fig 9B). However, uninduced *E. coli* can also absorb a certain amount of [α-$^{32}$P] ATP. We believe that this can be caused by the trace expression of PfUGT. Competition experiments with different potential substrates showed that the heterologously expressed PfUGT in *E. coli* were highly specific for ATP and ADP, with no competition from AMP, GTP, UTP, UDP-glucose, or UDP-galactose (putative substrate) with [α-$^{32}$P] ATP import. Therefore, we propose to name PfUGT as PfANT. However, PfANT also appears to have some uptake capacity for CTP. The ability of *E. coli* to take up radioactive ATP was inhibited in the presence of non-radioactive CTP, but the transport capacity for CTP lower than ATP and ADP (Table 2). Previous studies showed that a single amino acid mutation (F37V) in PfANT resulted in marked resistance to GNF179 and KAF156 [33]. Competition experiments showed

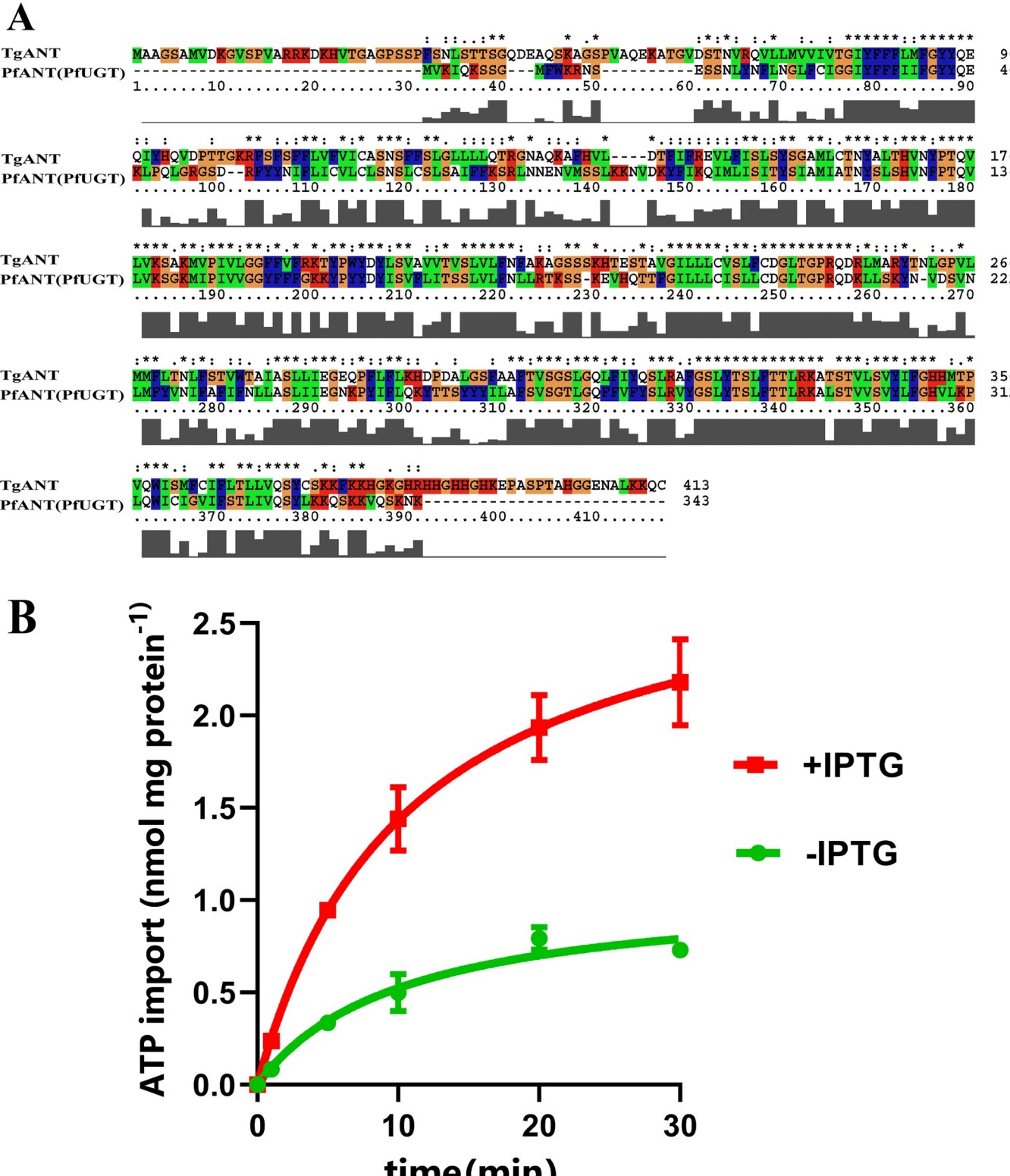

**Fig 9. Homology analysis and [α-³²P] ATP absorb experiment. (A)** The sequences of TgANT and PfUNT were aligned using ClustalX. **(B)** Kinetics of [α-³²P] ATP uptake into intact *E. coli* cells. IPTG-induced *E. coli* cells harboring the plasmid encoding PfANT were incubated with 25 nM [α-³²P] ATP for the indicated time intervals. Noninduced *E. coli* cells with the plasmid encoding PfANT were used as control. Means ± s.e.m., three independent assays.

**Table 2. Impact of potential substrates and effectors on ATP transport by PfANT.**

| Inhibitor | Uptake (% of control) | ± SEM (%) |
|---|---|---|
| **ADP** | **28.9** | **±3.8** |
| **ATP** | **23.8** | **±1.4** |
| **CTP** | **55.1** | **±7.3** |
| **GNF179** | **50.6** | **±2.9** |
| UDP-Gal | 89.2 | ±2.9 |
| UDP-Glu | 97.1 | ±3.9 |
| UTP | 97.6 | ±11.4 |
| AMP | 102.4 | ±5.2 |
| GTP | 95.0 | ±3.7 |

To test the substrate specificity of PfANT, a mixture of 25 nM [α-$^{32}$P] ATP and non-radioactive potential substrate at one-hundred-fold concentration was used to detect the inhibitory effect. *E. coli* cells expressing PfANT absorb of 25 nM of [α-$^{32}$P] ATP for 15 min were used as control. Means ± s.e.m., four independent assays.

that PfANT also has a certain transport capacity for GNF179, and its presence can inhibit the uptake of [α-$^{32}$P] ATP. Therefore, we speculate that the target of GNF179 is located in the ER of *Plasmodium*, and the mutation of PfANT leads to a decrease to absorb GNF179, resulting in drug resistance. This results provide a new direction for studying the resistance mechanism of GNF179.

## The depletion of TgANT leads to ER stress and apoptosis of *T. gondii*

One of the functions of ER ATP was to promote the correct folding of proteins and degrade the misfolded or overexpressed proteins. There is a strict protein quality control system in the ER, and this process relies on the ER chaperones and their cofactors. Based on this fact, the lack of TgANT is speculated to cause a sharp increase in the unfolded or misfolded proteins, which may lead to ER stress, also known as the unfolded protein response (UPR).

In mammals, the UPR is mediated by three types of proteins: kinase RNA-like ER kinase (PERK), activating transcription factor 6 (ATF6) and inositol-requiring enzyme 1 (IRE1) [23,34]. In apicomplexan parasites, amino acid deficiency, oxidative stress, accumulation of misfolded proteins in the ER can activate UPR. Unlike mammalian cells, the transcription factors IRE1, ATF6, and downstream XBP1, ATF4 that initiate UPR are absent in the genome of *T. gondii*. However, there are four eIF2α kinases (TgIF2K-A, TgIF2K-B, TgIF2K-C, and TgIF2K-D) to phosphorylate the α subunit of eukaryotic initiation factor 2 (TgIF2α) in *T. gondii* and direct translational and transcriptional modes of gene expression that regulate ER stress, oxidative stress, and the amino acid deficiency [18,35,36].

Thus, to determine whether the depletion of TgANT can give rise to ER stress and activate UPR, we examined the phosphorylation level of TgIF2α. The results showed that the depletion of TgANT caused a significant increase in the phosphorylation level of TgIF2α (Fig 10A). Previous research has shown that *T. gondii* must adapt to an extracellular environment while seeking a new host cell. Therefore, TgIF2α was phosphorylated after egress to initiate a parasitic stress response that promotes survival until a new host cell can be invaded [18,37]. To exclude the influence of extracellular tachyzoites, we collected intracellular and extracellular tachyzoites to detect their TgIF2α phosphorylation levels. Consistent with our expectations, the lack of the TgANT resulted in significant phosphorylation of TgIF2α in intracellular tachyzoites (Fig 10B).

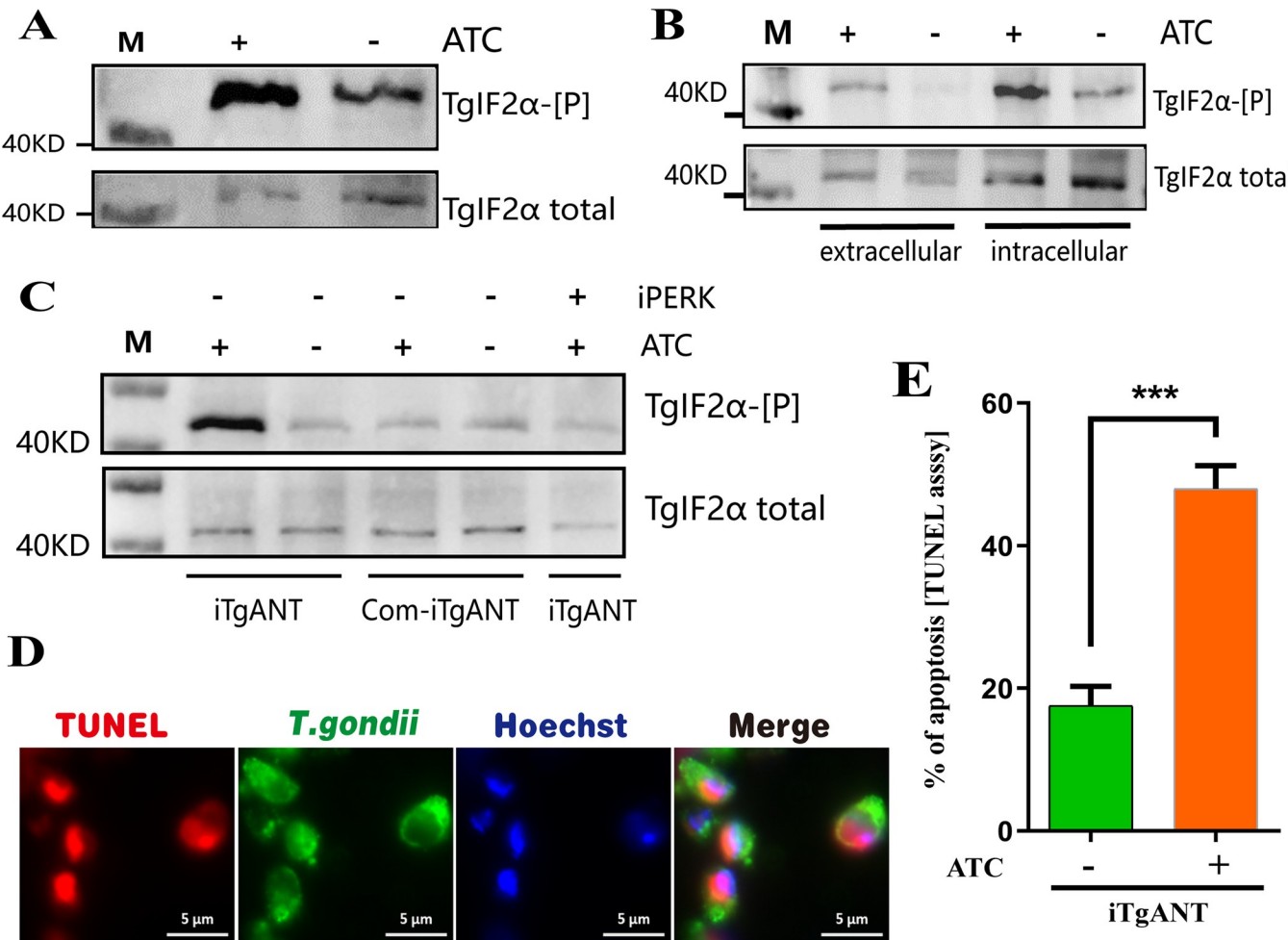

**Fig 10. ER stress and apoptosis detection. (A)** iTgANT strain were treated with 0.5 μg/ml ATc for 36h or left untreated, subsequently collected the parasites. They were subjected to Western blot analyses using mouse anti-TgIF2α-P and rabbit anti-TgIF2α. **(B)** Intracellular parasites were treated with 0.5 μg/ml ATc for 36h or left untreated, subsequently collected the extracellular and intracellular parasites. They were subjected to Western blot analyses using mouse anti-TgIF2α-P and rabbit anti-TgIF2α. **(C)** iTgANT strain and Com-iTgANT were treated with 0.5 μg/ml ATc for 36h or left untreated, subsequently collected the parasites. Besides, the iTgANT strain was treated with ATc for 33h and then add the iPERK for 3h to verify the phosphorylation of TgIF2α is caused by ER stress. All the samples were subjected to Western blot analyses using mouse anti-TgIF2α-P and rabbit anti-TgIF2α. **(D)** Morphology of iTgANT strain (green, GRA7) and TUNEL labeling in situ (red) following treatment with ATc for 36h. **(E)** Mean percentages ± SEM of TUNEL-positive *T. gondii*, results are from three independent experiments, ***P≤0.001, Student's t-test.

In apicomplexan parasites, amino acid deficiency, oxidative stress, accumulation of misfolded proteins in the ER can activate the phosphorylation of TgIF2α. Similar to PERK in mammals, TgIF2K-A localizes to the ER and is involved in sensing accumulation of misfolded and unfolded proteins in the ER through phosphorylation of TgIF2α [37,38]. Previous studies have shown that the inhibiter of TgIF2K-A (iPERK, GSK2656157) can only inhibit the phosphorylation of TgIF2α caused by ER stress [38]. In order to confirm that the phosphorylation of TgIF2α is caused by ER stress, we used iPERK to test whether it can reduce the phosphorylation of TgIF2α caused by the deletion of TgANT. As expected, iPERK treatment for 3 hours significantly reduced the phosphorylation level of TgIF2α (Fig 10C).

In mammals and plants, continuous ER stress can cause cell apoptosis [39]. Although most of the genes that have been identified as participating in the apoptosis pathway of metazoans are missing in the genomes of parasitic protozoans, such as ATF6, CHOP, and XBP, the

signature feature of apoptosis were observed and identified in the protozoa, including shrinkage and rounding of the body shape, DNA fragmentation, and nucleus condensation[40–42]. Besides, a few characterizations of apoptosis have been identified in *T. gondii* treated with apoptosis inducers [43]. We believe that the depletion of TgANT leads to the interruption of ATP supply in the ER, and the loss of ATP causes irreversible ER stress, which in turn results in *T. gondii* apoptosis. DNA fragmentation is one of the common downstream biochemical markers of cell apoptosis in mammals and is also considered to be an important marker of protozoan apoptosis-like cell death[43,44]. Therefore, in the next step we detected the DNA fragmentation of the TgANT depletion strain by TdT-mediated dUTP Nick-End Labeling. The result showed that a significant increase in DNA fragmentation was observed in the iTgANT strain treated with ATc for 36 hours (Fig 10D and 10E). In conclusion, the depletion of TgANT is associated with apoptosis of *T.gondii*, and may be one of the causes of death.

## Discussion

Taken together, we identified TgANT as an ATP/ADP transporter located in the ER of *T. gondii* by radioactively labeled [α-$^{32}$P] ATP and other methods. This is the first-time proof that a specialized transporter is required to transport ATP in the ER of the apicomplexan parasite to ensure the normal ER function. Our results indicate that TgANT is essential for the normal growth of *T. gondii*. We also found that the two conserved domains of TgANT may be the ATP binding sites, and the *E. coli* expressing the mutant plasmid can no longer absorb ATP. As expected, in our study mutations in these two conserved domains make the ectopic expression of TgANT unable to rescue the growth defect of *T. gondii*. The genome of several organisms including yeast, plants and mammals contains multiple genes encoding the mitochondria ADP/ATP transporter, and have high homology [45–48]. To determine whether there are multiple genes encoding the ER ADP/ATP transporter of *T. gondii*, we used protein sequence of TgANT to find whether there is a homologous protein after we verified the function of TgANT. But we did not find another homologous protein.

Like AXER in mammals, TgANT was also considered as a UDP-galactose transporter in the database before. Experiments have shown that AXER and TgANT do not have the transport function of UDP-galactose. Besides, the phenotype score associated with the TgANT encoding gene (−4.9) was significantly lower than most genes in *ToxoDB*, suggesting that TgANT may have much more severe effects in *T.gondii*. This is also consistent with the result that TgANT does not have UDP-galactose transport capacity because the lack of UDP-galactose transporter in Lec8 cells does not significantly impact cells growth. Previous studies had suggested that glycosylation is essential in *T. gondii*, various glycosylation types and many glycosylation-related enzymes have been identified in *T. gondii*. However, there are few studies on glycosylation substrate transporters, many transporters conserved in animals and plants have been found no homology with the proteins in *T. gondii* [49,50]. The glycosylated substrate transporters currently annotated in *ToxoDB* also have high CRISPR phenotype values, indicating that glycosylated substrates may have other transport mechanisms in *T.gondii*. If TgANT is only the UDP-galactose transporter, then theoretically, the lack of TgANT would likely not cause the fatal growth defect of *T.gondii*.

At present, there are only preliminary studies performed on the ER ATP/ADP transporter. Interestingly, knockdown AXER does not cause obvious growth defects in HeLa cells, which is very different from the severe growth defects observed in *Arabidopsis* and *T.gondii*. This suggests that there may be other ATP transport mechanisms in the ER of HeLa cells. It may also be because HeLa cells are not specialized secretory cells. Knockdown of AXER has no fatal effect. Of course, it may also be because siRNA cannot eliminate the protein expression ability

of AXER. The analysis showed that siRNA knocked down the residual AXER level to 10% ~20% [14]. Therefore, ATP transported by deficient protein expression may be sufficient to maintain the normal growth of HeLa cells. In fact, the TATi inducible knockdown system used in this experiment cannot eliminate the expression of TgANT at the protein level. These results indicate that for *T.gondii* with specialized secretory organelles, a large amount of ATP is required in the ER as an energy source to maintain the normal secretion of secreted proteins, thereby maintaining the normal growth of *T. gondii*. In addition, we found that two key domains are essential to the function of TgANT. The *E. coli* expressing the mutant plasmid can no longer absorb ATP, and the mutation of these two domains cannot restore the growth defect of the iTgANT strain. These results once again confirmed our speculation that the phenotypic defect caused by the depletion of TgANT is mainly caused by the lack of ATP in the ER of *T. gondii*. Since these two highly conserved domains exist in *T. gondii*, *Plasmodium*, *HeLa cells*, and *C.elegans*, We speculate that this transport mechanism may be applicable in multiple species. We concede that it was more convincing if we could directly measure the ATP content in the ER. We tried to express ATP FRET sensor ERAT4.01 [51] in the ER of *T. gondii* to directly measure the ATP, but a variety of guide sequences could not guide FRET sensor ERAT4.01 to the ER. Current studies have shown that the mutation of a single amino acid in the human mitochondrial adenine nucleotide translocator (ANT1) leads to the loss of ATP transportability and causes serious diseases [52]. Since we mutated four amino acids in these two protein domains, we do not know which one or which of these amino acids was the most critical for TgANT. This may be what we need to further improve next.

The ER is an important place for the processing and folding of secreted proteins and membrane-bound proteins. The ER has a strong demand for ATP, especially secretory cells with professional secretion ability [51,53]. We believe that the decrease of ATP will lead to a sharp increase of unfolded proteins in the ER in *T. gondii*, which causes ER stress. Accordingly, we detected the phosphorylation of the transcription initiation factor TgIF2α, which plays a primary regulatory function during ER stress [37,38,54]. As the result, the lack of the TgANT significantly enhanced the phosphorylation level of TgIF2α. This mode of regulation is fundamental in ER stress, which can reduce proteins that enter the ER and preferentially express molecular chaperones and other related proteins. Interestingly, similar phenomena were also observed in *Caenorhabditis elegans*. The hut-1 (SLC35B1 family gene: a homolog of UDP-Gal transporter, Located in the ER) deletion mutant and RNAi worms induced chronic ER stress and showed larval growth defect and lethality with disrupted intestinal morphology in *Caenorhabditis elegans* [15]. However, lethality and the ER stress phenotype of the mutant were rescued with the human hut-1 ortholog UGTrel1 (AXER). This indicates that hut-1 is likely to be the ATP/ADP transporter in the ER of *C. elegans*. Besides, previous studies have shown that expression of the SLC35B1 gene was increased under ER stress conditions in various organisms, such as *Caenorhabditis elegans* [55], mouse embryonic fibroblasts [32], and cultured human cells [56]. However, the ER stress caused by a deficiency of the ER ATP/ADP transporter does not seem to be universal. Mutant plants lacking ER-ANT1 exhibit a severe dwarf phenotype but do not cause apparent ER stress in *Arabidopsis thaliana*. The absence of ER-ANT1 activity mainly affects photorespiration (maybe solely glycine decarboxylase) and primary cellular metabolism remains unchanged. Meanwhile, depletion of AXER from HeLa cells only reduces the ER ATP levels and BiP activity, and there is no ER stress and severe growth defects.

Unlike the ER ATP/ADP transporter in mammals, apicomplexan parasites and nematodes, ER-ANT1 in plants has high homology with the mitochondrial ATP/ADP transporter (AAC). Interestingly, all these effects observed with the ER-ANT1 mutants can be reverted to a wild-type-like phenotype by plant growth at a high external $CO_2$ concentration. The author considers that the absence of ER-ANT1 leads to ROS accumulation. Nevertheless, why ER-ANT1

causes the accumulation of ROS is still unknown, which indicates ER-ANT1 may have multiple functions in Arabidopsis [13].

Our data proved that PfANT in *P. falciparum* is also an ATP/ADP transporter, not a UDP-galactose transporter. Interestingly, PfANT not only able to transport ATP and ADP, but also has a certain ability to transport CTP and the antimalarial drug GNF179. Previous studies have suggest that the mutation of one amino acid (F37V) of PfANT resulted in resistance to two antimalarial imidazolopiperazines, KAF156 and GNF179, and the mechanism of resistance is not clear [33]. In *P. falciparum*, although the mutate of PfANT (F37V) lead to resistance to GNF179 and KAF156, but this mutation also makes it difficult to survive in animals. Parasites that harbor mutate of PfANT proliferate at a reduced rate. The researchers speculate that these low fitness mutants might not survive the competition for nutrients in a polyclonal environment such as the human body, or may be more readily cleared by the immune system [33]. However, we speculate that PfANT mutants may have obstacles to the transport of substrates (ATP, ADP or CTP), thus resulting in the inhibition of mutant growth in vivo. The sequence before and after the thirty-seventh amino acid (F) in *P. falciparum* is also conserved in *T. gondii*. We tried to express the mutated TgANT (F81V, corresponding site in PfANT is F37V) at the *UPRT* locus, however, it has never been obtain the mutant strain.If we can obtain the gene mutant strain in *T. gondii*, it will be a suitable material for investigating the mechanism of drug resistance in apicomplexan parasites. To conclude, TgANT is a new type of ATP/ADP transporter in *T. gondii*. Through the functional identification of TgANT, the physiological importance of energy supply in ER of *T. gondii* is revealed for the first time. As an obligate intracellular parasitic protozoan, *T. gondii* has a variety of specialized organelles of protein secretion, many proteins are processed and folded in the ER, which requires a large amount of energy. The depletion of TgANT likely leads to a lack of ATP in the ER, which may explain why the iTgANT strain has serious growth defects.

## Materials and methods

### Ethics statement

The preparation of polyclonal antibody against TgANT (*T. gondii*) based on the principles of welfare and ethics of experimental animals. The project optimized the design scheme and strictly planned the number of animals needed. All animal experiments were approved by the ethical committee of Huazhong Agricultural University. (ID Number): HZAUMO-2020-0077

### Parasite and cell culture

All strains and plasmids used in this study are listed in S2 Table. Strain RH△ku80, RH△hxgprt (RH△hx), TATi of *T.gondii* was cultured on monolayers of human foreskin fibroblasts (HFF) with Complete Medium, which consisted of DMEM (Dulbecco's modified Eagle's medium) and supplemented with 2% or 10% (v/v) fetal bovine serum (FBS), 2 mM L-glutamine, 100 units ml$^{-1}$ penicillin/streptomycin (Corning) at 37˚C and 5% $CO_2$) incubator. Chinese hamster ovary cell lines Lec8 is a mutant clone derived from the parental CHO and deficient in UDP-Gal transport, which was purchased from the American Type Culture Collection and with 10% FBS, 100 units ml$^{-1}$ penicillin/streptomycin (Corning), 2 mM glutamine, proline 40 mg L$^{-1}$ at 37˚C in an atmosphere of 5% $CO_2$.

### Plasmid and mutant strains construction

All primers used in this study are listed in S3 Table in the supplemental material, and the plasmids were constructed by multi fragment ligation using the ClonExpress II one-step cloning kit (Vazyme Biotech).

CRISPR/cas9 plasmids were used to construct the different mutant strains, and all the CRISPR/cas9 plasmids were generated by replacing the *UPRT* targeting guide RNA (gRNA) in pSAG1-Cas9-sg*UPRT* with corresponding gRNAs [28,57].

The iTgANT conditional expression mutant containing an N-terminal Ty tag (EVHTNQDPLD) was generated by amplifying the CDS of TgANT to replace the PYK1 cassette in pSAG1-TetO7-PYK1 [28]. The resulting construct was linearized by upstream and downstream primers, transfected into the TATi strain with 7.5 μg CRISPR/cas9 plasmids, and allowed to infect fresh HFFs for 24 hours post-transfection, clones isolated by limiting dilution using 1 μM pyrimethamine. Anhydrotetracycline (ATc,TaKaRa) was used to deplete TgANT expression in all iTgANT strains at a final concentration of 0.5 μg/ml [22,57,58]. All the complementation and heterologous complementation strains were generated in the *UPRT* locus using pSAG1-Cas9- sg*UPRT* plasmid as previously described [57,58].

For the complementation assays in mutant CHO Lec8 cells, the pCDNA-3.1 universal plasmid was used for generating the plasmid containing TgANT-CDS (pCDNA-3.1-TgANT-HA) and hUGT2-CDS (pCDNA-3.1-hUGT2-HA). These two plasmids containing an N-terminal HA-tag were generated by PCR amplifying the CDS form of *T.gondii* cDNA and HFF cDNA. Each CDS fragment has a homology arm inserted into the multiple cloning site (MCS) of the pCDNA-3.1 expression vector.

For the construction of iTgANT::AXER heterologous supplementation, we obtained the nucleotide sequence of the C-terminal 63 amino acids of the TgANT and the CDS of AXER (SLC35B1, Fig 1A) using specific primers. Using the method of homologous recombination, the nucleotides corresponding to the 63 amino acids at the C-terminus of TgANT were connected to the C-terminal of AXER as a guide sequence (fusion AXER protein sequence were in the S4 Table). Followed, the fusion AXER protein were cloned into the pUPRT-CAT as previously described [28].

## Heterologous expression in *E. coli*

To improve the expression of TgANT in *E. coli*, codon optimization was performed, and the Optimized Sequence was shown in S5 Table. The wild-type Optimized Sequence and mutant CDS of TgANT were cloned into the *E. coli* expression vector pET16b, and the inserts were confirmed by sequencing. Transformations of *E. coli* were carried out according to standard protocols, and the BL21 (DE3) strain was used for heterologous expression. *E. coli* cells transformed with the TgANT expression plasmids were grown at 37˚C under aerobic conditions in TB$^{Amp/Clm}$ medium (TB: 2.5 g/L KH$_2$PO4, 12.5 g/L K$_2$HPO$_4$, 12 g/L peptone, 24 g/L yeast extract, 0.4% glycerin, pH 7.0) [20]. To induce heterologous protein synthesis, an *OD600* = 0.5–0.6 was started the initiation of T7-RNA polymerase expression by the addition of 1 mM IPTG. In addition, an equal amount of *E. coli* without IPTG induction was used as control group. Cells were grown for 1.5 h after induction or uninduction and collected by centrifugation for 2 min at 7000g. The cells were resuspended to *OD600* = 5 using potassium phosphate buffer (50 mM, pH 7.0).

## Uptake of radioactively labeled nucleotide

Take 100ul of IPTG-induced and uninduced *E. coli* cells (*OD600* = 5) and discard the supernatant after centrifugation, then resuspend the cells pellet with 100ul potassium phosphate buffer containing radioactively labeled [α-$^{32}$P] ATP. The uptake of nucleotides was carried out in an Eppendorf reaction vessel at 30˚C [20]. After the specified time, the reaction was terminated by adding 400ul cold potassium phosphate buffer. Cells were further centrifuged and washed to remove unimported radioactivity ATP by the addition of 400ul potassium phosphate buffer.

Following the cells, the pellet was resuspended with 50ul potassium phosphate buffer and 100ul ULTIMA Gold (PerkinElmer) in a Clear 96-well Flexible PET Microplate (PerkinElmer, 1450–401). Radioactivity in the samples was quantified in Perkin Elmer MicroBeta TriLux (1450LSC).

## Thin-layer chromatography of radioactively labeled adenine nucleotides

Induced *E. coli* cells harboring TgANT were incubated with 50 nM radioactively labeled [$\alpha$-$^{32}$P] ATP for 5 minutes, non-imported radioactivity was removed by washing (five times in PBS). Subsequently, *E. coli* cells were incubated in PBS with 5 µM non-labeled ATP, ADP, or PBS for 3 minutes and terminated by rapid centrifugation. Two microliter radioactively labeled sample of the supernatant were loaded onto the TLC PEI cellulose plate (MERCK), and dried with a fan. Separation was carried out for 1 M ammonium sulfate and 1.5 M mono-potassium phosphate. Radioactively labeled [$\alpha$-$^{32}$P] ATP and [$\alpha$-$^{32}$P] ADP were determined after autoradiography and visualized under UV light. The position of exported label was identified by comparison with standard of radioactively-labeled [$\alpha$-$^{32}$P] ATP.

## Plaque assay

Plaque assays were performed using freshly confluent 6-well plates of HFFs infected with 200 parasites per well in the presence or absence of ATc and incubated at 37˚C without disturbing for seven days. The plaques in HFFs were fixed with methanol and stained with a crystal violet solution, and the total number of plaques were observed and counted using photoshop software. All plaque assays were performed in biological triplicate.

## Lectin and western blotting

For lectin blot, T25 flask infected with wild-type strains and iTgANT strains in the presence of 0. 5µg/ml ATc were incubated 36 hours at 37˚C in an atmosphere of 5% $CO_2$, and then the parasites were released from syringe-lysed HEFs and 3µm filter membrane were used to filter and collect the parasites. Parasites were collected and lysed using RIPA (Beyotime, supplemented with protease inhibitor cocktail and EDTA) according to the manufacturer's instructions, and were boiled for 10 min with SDS Loading Buffer.

For western blot, the sample processing method was the same as above. Aliquots containing 20 µg of total proteins samples were loaded and separated by precast 10% SDS-PAGE gels. After electrophoresis, proteins were transferred to a nitrocellulose membrane, and unoccupied protein binding sites on the membrane were blocked by incubation with 1% BSA in TBS supplemented with 1% Tween-20 (TBST) for two hours. For detection of the galactosylation proteins, membranes were blotted with lectin GS-II Alexa Fluor 488 conjugate (40 µg mL$^{-1}$, 1:1000 dilution, Thermo, L21415)) for two hours at room temperature with gentle agitation. For detection of the Ty and HA epitope, TgIF2$\alpha$ and TgIF2$\alpha$-P, membranes were incubated overnight with a 1:1000 dilution primary antibody at 4˚C, followed by incubation with a 1:1000 dilution of HRP-conjugated secondary antibodies. Immunoreactive bands and lectins bound were visualized by ECL detection Western Lightning Chemiluminescence Reagent Plus system.

## Immunofluorescence assay

For immunofluorescence analysis, HFF monolayers on the coverslip (24-well plate) were infected with 50µl of the fresh parasite strains that were released from syringe-lysed HFFs, after 45 minutes, uninvaded parasites were washed off with PBS. Twenty-four hours after

infection, coverslips were fixed with 4% paraformaldehyde for 20 minutes and permeabilized with 0.1% Triton X-100 in PBS for 20 minutes. Primary antibodies and secondary antibodies diluted in 10% BSA-PBS were incubated for 16 minutes at 37°C, and coverslips were washed five times with PBS after primary antibodies and secondary antibodies were incubated. Hoechst was used as a co-stain to visualize host and parasite nuclei. Coverslips were mounted using Vectashield antifade mounting medium. For immunofluorescence analysis the expression of TgANT in *E. coli* was performed as described in Seongjin Park [59].

## Invasion assay

This protocol was adapted from Albuquerque-Wendt A and Bandini G [60,61]. Firstly, iTgANT strains were treated with ATc or not for 36 hours, and were released from syringe-lysed HFFs and centrifuged for 5 min at 1000 g. HFF monolayers grown on coverslips in 24-well plates were infected with $10^5$ tachyzoites/well, and parasites were allowed to invade cells for 15–20 mins before fixation with 4% paraformaldehyde. Extracellular parasites (green) were stained using pig anti-*T. gondii* serum (1:1000) and FITC-conjugated goat anti-swine IgG in nonpermeabilized conditions. Cells were then permeabilized (20 min, 0.1% Triton X-100), and all parasites (extracellular and intracellular) were stained with rabbit polyclonal anti-TgALD (1:1000) primary antibody and followed by incubation with IgG Alexa Fluor 594. The nucleus of HFF cells and parasites was stained with Hoechst. Randomly selected fields (15–20) were counted and the percentage of intracellular parasites was calculated. Invasion efficiency was determined as the number of invaded (red-green) parasites divided by the number of the hose cell nucleus.

## Antibody

The antibodies used in this experiment included rat/mouse anti-HA (MBL, 1:1000 or 1:500), mouse anti-Ty (Prepared by our laboratory, 1:1000 or 1:500), goat anti-rabbit/rat Alexa Fluor 594 (Invitrogen, 1:1000), goat anti-rabbit/ rat Alexa Fluor 488(Invitrogen, 1:1000), rabbit anti-mouse Alexa Fluor 594 (Invitrogen, 1:1000), rabbit anti-mouse Alexa Fluor 488 (Invitrogen, 1:1000) and hochest 33342 (Beyotime, 1:1000), mouse anti-TgIF2α-P (HUABIO, 1:1000), rabbit anti-TgIF2α (ABclonal, 1:1000), goat anti-rabbit/rat HRP (Beyotime, 1:1000), rabbit anti-Ompf (Biorbyt, 1:500), rabbit anti-ALD (TGGT1_236040; Prepared by our laboratory, 1:1000) [62], rabbit anti-SERCA (Prepared by our laboratory, TGGT1_230420, 1:500) [63], rabbit anti-GAP45 (Prepared by our laboratory, 1:500), rabbit anti-IMC (Donated by Professor Qun Liu of China Agricultural University) [64].

## Supporting information

**S1 Fig. Location and lectin experiment.** (A) To determine the subcellular location of *TGGT1_273390*, *TGGT1_300360* and *TGGT1_249900* in *T.gondii*, 3×HA-tagged was cloned into the C-terminus by CRISPR-Cas9–mediated site-specific integration in the RHΔku80 strain. IFA confirmed the subcellular location of *TGGT1_273390*, *TGGT1_300360*, *TGGT1_249900*. Hsp60(Heat shock protein 60) as a marker of mitochondrial. Scale bars: 2 μm. (B) IFA staining to determine the subcellular location of TgANT-10×HA strain.(C) Lec8 cells were transfected with pcDNA3.1- TgANT-HA or with pcDNA3.1- hUGT2-HA, followed cells were examined for the binding of GS-II-488 (green) and anti-HA monoclonal antibody (red). Bar, 20μm. (D) The iTgANT strain was treated with 0.5 μg/ml ATc or left untreated, then pipetted with the syringe and filtered to collect the tachyzoite. Lectin GS-II-488 was used to detect the galactosylation level of these two samples by lectin blot.
(TIF)

**S2 Fig. The Phenotypic experiment of heterologous supplementation strains.** (A) IFA confirmed the correct integration and expression in ER of iTgANT::hUGT2 strain. (B) and (C) Intracellular replication assay(B) and plaque assay(C) comparing the growth of TgANT depletion strain and hUGT2 complementation strain. Means ± s.e.m. ***p ≤ 0.001, two-way ANOVA, three independent repeated, a representative one is shown here.
(TIF)

**S1 Table. Potential transporters screened out from *ToxoDB*.** In order to narrow the screening range, we set the transmembrane domain greater than 6 (≥ 6), and the CRISPR phenotype value less than -3 (≤ -3) as our screening threshold.
(XLSX)

**S2 Table. Strains and plasmids used in this study.**
(DOCX)

**S3 Table. Primers used in this study.**
(XLSX)

**S4 Table. The amino acid sequence of AXER fusion protein.**
(DOC)

**S5 Table. The codon optimization scheme of TgANT.**
(DOC)

## Author Contributions

**Conceptualization:** Jiahui Qian, Junlong Zhao, Rui Fang.

**Data curation:** Senyang Li.

**Formal analysis:** Senyang Li.

**Funding acquisition:** Rui Fang.

**Investigation:** Senyang Li.

**Methodology:** Senyang Li, Jing Yang.

**Project administration:** Senyang Li, Jiahui Qian, Junlong Zhao.

**Resources:** Zhengming He.

**Software:** Senyang Li, Ming Xu, Tongjie Zhao.

**Writing – original draft:** Senyang Li.

**Writing – review & editing:** Senyang Li.

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
