## [Decision Letter · Decision Letter 0]

4 Mar 2022

Dear Dr. Fang,

Thank you very much for submitting your manuscript "A new adenine nucleotide transporter located in the ER is essential for maintaining the growth of Toxoplasma gondii" for consideration at PLOS Pathogens. As with all papers reviewed by the journal, your manuscript was reviewed by members of the editorial board and by several independent reviewers. In light of the reviews (below this email), we would like to invite the resubmission of a significantly-revised version that takes into account the reviewers' comments.

The reviewers were overall enthusiastic about the novelty and significance of this study characterizing the first ER-localized ATP/ADP transporter in T. gondii. As noted by the reviewers, additional data and controls are needed to substantiate certain experimental findings. Additionally, please note the need to provide additional descriptions for some experiments (e.g. sources of reagents, sequences of expression constructs, and more detailed labeling of figures). Please also provide a thorough description of the number and type of replicates used for all experimental findings and the statistical methods used.

We cannot make any decision about publication until we have seen the revised manuscript and your response to the reviewers' comments. Your revised manuscript is also likely to be sent to reviewers for further evaluation.

Sincerely,

Sean T Prigge

Guest Editor

PLOS Pathogens

Dominique Soldati-Favre

Section Editor

PLOS Pathogens

Kasturi Haldar

Editor-in-Chief

PLOS Pathogens

orcid.org/0000-0001-5065-158X

Michael Malim

Editor-in-Chief

PLOS Pathogens

orcid.org/0000-0002-7699-2064

The reviewers were overall enthusiastic about the novelty and significance of this study characterizing the first ER-localized ATP/ADP transporter in T. gondii. As noted by the reviewers, additional data and controls are needed to substantiate certain experimental findings. Additionally, please note the need to provide additional descriptions for some experiments (e.g. sources of reagents, sequences of expression constructs, and more detailed labeling of figures). Please also provide a thorough description of the number and type of replicates used for all experimental findings and the statistical methods used.

Reviewer's Responses to Questions

**Part I - Summary**

Reviewer #1: This is a very interesting study, showing that the T. gondii UGT family protein TGGT1_254480 is important for the growth of T. gondii, and examining the consequences of its depletion on ER processes. It provides multiple lines of evidence that, despite its annotation, the protein is not a UDP-galactose transporter. Data consistent with an ER localisation and an ATP/ADP exchange function are presented. The study is novel and significant. Including data for PfUGT, which are eluded to in the text but not shown, would increase its significance further. There are some issues in the manuscript that require clarification and additional details that are needed. Furthermore, the transport studies would benefit from additional controls, and the localisation studies from additional markers.

Reviewer #2: In this manuscript, the authors characterized a Toxoplasma ortholog of ADP/ATP transporter. To determine the substrates of TgANT, the authors cloned this gene in E. coli and identified its capability of importing ATP. In addition, the authors complemented the TgANT KO with human UGT2 and did not observe complementation on growth. However, the complementation with the AXER gene rescued the growth. These assays presented solid evidence that the TgANT transports ATP/ADP and is essential for parasite growth. But, the study lacks mechanistic characterization of how the deletion of TgANT affects ER physiology, further causing parasite death.

Reviewer #3: In this manuscript, the authors describe the identification of the first ER-located protozoan ATP-ADP exchanger in a protozoan species. This is a commendable study, breaking new ground in an important aspect of the cell biology of protozoa, and specifically, apicomplexan parasites. I find the results clearly presented and the conclusions well sustained by the presented evidence, with few exceptions. The manuscript is clearly written and structured and easy to follow, although there are many small issues with written English that need attention, in my opinion.

I have no major comments to make on the scientific validity of this work but do have a number of comments and suggestions for the improvement of the paper. Most of these are minor.

**Part II – Major Issues: Key Experiments Required for Acceptance**

Reviewer #1: The following additions would strengthen the study, but may not be absolutely required for acceptance:

1. additional controls in the E. coli transport experiments

2. a plasma membrane marker in the localisation studies

3. the data for the function of PfUGT

These suggestions are outlined in more detail in the next section.

Reviewer #2: 1. Line 124-137, TGGT1_273390, TGGT1_300360, and TGGT1_249900 are highly homologous to ER-ANT1; why are they excluded from being considered as putative TgANT1. At the end of this paragraph, please conclude which gene is named TgANT1 for clarity.

2. Line 138-140, please explain why the peptide fragment outside transmembrane regions of TgANT1 is not available. Can it not be predicted by topology prediction software?

3. For Figure 3A, in the presence of unlabeled ATP and ADP, the 32P-ATP-preloaded E. coli transports out both ATP and ADP, but the ratio seems the same regardless of the addition of ATP or ADP extracellularly. In the presence of ATP, ADP will be transported out more than ATP and vice versa. Please explain why no difference is observed between Lane 2 and 3.

4. Fig 3B. Controls without the addition of ATP and ADP are needed as a reference.

5. Line 234-236, why does the N-terminally HA-tagged TgANT fail to be detected by WB? The tag facing the ER lumen is degraded? If so, why did IFA work?

6. Fig 4C. It should be cautious about stating that the expression of TgANT is completely abolished at 24-h post-ATc treatment. There are still some stainings of HA by IFA.

7. Fig 6-8 showed strong evidence that a conserved peptide motif is key for TgANT function. The author claimed that the TgANT transports ATP into the ER lumen for protein biosynthesis. However, there is a lack of in vivo evidence to support this critical information from this manuscript. I suggest that the authors consider introducing an ATP/ADP biosensor to quantitatively compare the ER's energy status between WT and KO, such as ATP level and ATP/ADP ratio, or use other biochemical strategies. You can refer to the following paper for more information about biosensors. https://www.ncbi.nlm.nih.gov/pmc/articles/PMC3852917/

8. For Fig 9, in panel A-C, the translation is arrested in the TgANT knockout via upregulated phosphorylation of TgIF2a. Since the replication is reduced dramatically in the KO, it is not surprising to see these phenotypes. There are a few possibilities that ER stress can cause, such as protein folding defects, abnormal ER-associated protein degradation, and ER stress-induced apoptosis. The author quantified higher apoptosis in the KO than WT. Due to the upregulation of eIF2a, the Ca2+ concentration in the ER should be decreased, and the ER-mitochondria contact sites get increased. Further, the Ca2+ is transferred to mitochondria, which induces apoptosis. The authors need to test this hypothesis in the TgANK KO to understand the molecular mechanism by which the deletion of TgANT triggers apoptosis.

Reviewer #3: (No Response)

**Part III – Minor Issues: Editorial and Data Presentation Modifications**

Reviewer #1: ER localisation.

Lines 140-150: The authors describe two different tagging strategies that were used to localise TgANT, in two different strains. It is not clear why the authors do not show the data for both strategies.

Fig. 1C: A plasma membrane marker and a DIC image would be helpful to demonstrate the ER localisation of TgANT.

The antibody used to label SERCA is not named and its source is not provided. It is not stated whether this antibody is specific to the SERCA from T. gondii or not. These issues should be fixed.

Transport studies.

Information should be provided on whether there were an equal number of viable bacteria in the IPTG-induced and non-induced conditions in the ATP transport experiments.

It would be useful to test an additional radiolabeled substrate as a control – one that is not a substrate of TgANT but that is taken up by bacteria (to check that IPTG induction/TgANT expression does not non-specifically increase the uptake of unrelated compounds).

The Michaelis-Menten curves from which the Km and Vmax values were derived should be shown. As it currently stands, a Km and Vmax are cited in the text, with no error and no detail anywhere in the manuscript about the conduct of the experiments from which these values were derived.

Does ‘three-time repetition’ (used numerous times throughout the manuscript) refer to technical replicates or independent experiments performed on different days? This should be clarified. Independent experiments are required for the transport studies.

PfUGT.

The authors state that PfUGT also functions as an ATP/ADP transporter but do not show the data. In the Discussion, they state: “our data proved that PfUGT (PfANT) in P. falciparum is also an ATP/ADP transporter”. Perhaps the authors plan to submit a separate study on PfUGT, in which case this would be good to clarify. Otherwise, it is not clear why the authors have chosen not to present these very important data that would be of great interest to the malaria field. With a functional assay for PfUGT, it would also be interesting to test whether ganaplacide affects the function of the protein.

Clarity and additional details.

Overall, the manuscript needs editing to improve clarity.

Line 182: should Fig S2A be Fig S1A?

Fig 3A: it would be helpful to state what is in each of the lanes on the Figure itself or in the Figure legend.

Fig 4C: what is ALD? This should be explained in the legend, and the source of all the antibodies used in the study, including the anti-ALD antibody, should be provided.

Was iTgANT1 made in a Delta-ku80 background (i.e. is the strain the authors call ‘TATi’ TATi-Delta-ku80? The sources/references for the parent strains used in the study should be provided.

The titles of Figures 6 and 7 need changing as the E. coli uptake experiments are shown in Figure 6 not Figure 7.

Line 587 – should this be F37V (the resistance-conferring mutation in PfUGT)? The relevant studies about its role in resistance should be cited. There are no references at all in the relevant paragraph of the Discussion, which should also include references to the statements pertaining to Arabidopsis.

Lines 591 – 592: editing is required, the meaning is unclear.

It is not clear exactly how the ER-localising AXER fusion protein was made – more detail, including exactly which residues of both AXER and TgANT were present in the fusion protein, is needed.

Fig 9B: it is not clear what was loaded into which lanes (e.g. what is the difference between lanes 2 vs 4 and 3 vs 5 – proteins from intracellular vs extracellular parasites?

Reviewer #2: 1. The authors gave a comprehensive literature review for ER ANT and AXER but barely mentioned the ER stress studies in Toxoplasma and other pathogens. It would be great to see this information in the Introduction.

2. Line 141, a typo, C-C-terminus

3. Line 296, a typo, potential

Reviewer #3: Abstract, line 16: ‘lumens of the endoplasmic reticulum (ER) is the subcellular sites’ should be lumen and should be site (not the plurals). There are many such minor grammatical issues, which of course I will not all highlight.

In the abstract, line 22, please introduce the acronym TgANT1 after ‘an ATP/ADP transporter’.

Line 41: ‘ADP/ADP transporter(TgANT)’ should be ATP/ADP transporter. Also, throughout the manuscript, space before opening brackets is missing. Could be an issue with the software.

Line 58: ‘required/needed’ either word should be fine.

Line 92-93: ‘it only caused the activity of the BIP protein to decrease and slight cell growth slowed down.’ The BIP protein is not introduced and I do not understand this statement.

Line 98-99: Caenorhabditis elegans in italics as species name please. (also line 349)

Line 115-116: ‘we heterologous expression TgANT in E. coli, this system has previously…’.Multiple grammatical errors in this sentence.

Line 124: ‘Blastp analysis revealed only three sequences encoding putative solute carriers family protein’. I do not think this can be correct as stated. However, it is quite possible that BLASTP analysis identified only three genes (within the set parameters), which were putative solute carriers.

Line 130. ‘experiments show that they are located in the mitochondria instead of the ER.’ This needs clarification and perhaps some references. Which experiments, described where?

Line 136: ‘C-terminally dilysine motif shown in other proteins to limit exit from the ER (Fig 1B).’ I fail to see this motive in Figure 1B. It is possible Figure 1A is meant instead, but the reproduction of the small font size is such (in my copy) that I can’t make it out.

Line 139-140 ‘…but it was not available’. Sorry, this is not clear. You tried to prepare an antibody but it was not available? Do you mean ‘not successful’? Or the epitope was not available for binding of antibodies? Please clarify.

Line 141: I am not familiar with the description C-C terminus.

Line 150: co-localized, not ‘co-localization’.

Figure 1 legend: ‘The nitrogen termini and carboxy termini face the ER lumen, the double lysine motif(-KKQC) near the C-terminally of TgANT’ should be ‘The nitrogen terminus and carboxy terminus face the ER lumen, the double lysine motif (-KKQC) is located near the C-terminal end of TgANT.’

Line 168-170: ‘Western blot and IFA. As expected, the TgANT was expressed and inserted into the bacterial membrane(Fig 2A).’ Is this Western blot shown anywhere?

Line 172-175: ‘The import of radioactively labelled [α-32P]ATP by the recombinant TgANT into E. coli cells displayed typical Michaelis-Menten kinetics, with apparent Km values of 3.719 μM and Vmax of 6.974 nmol/min·mg for ATP.’ The determination of the Km is important but the data are not shown anywhere. Is this a single determination? If the average of multiple experiment, SD or SEM should be given.

Line 175-176: ‘the uninduced E. coli was not able to absorb ATP or ADP at significant rates (Fig 2B).’ Actually the slope of the line in Figure 2B looks non-zero, probably a significant uptake albeit of course very much lower than the cells expressing TgANT1.

Line 179: ‘The ability of E. coli to uptake radioactive ATP..’ take up, not uptake (uptake is not a verb).

Line 184-185: ‘PfUGT also does not have the UDP-galactose transport capacity but has a strong affinity for ATP and ADP(Data not shown).’ Sorry but this is an important claim and requires the data to be shown - or the claim not made - as far as I am concerned. Same for lines 585-586 in the Discussion: ‘our data proved that PfUGT (PfANT) in P. falciparum is also an ATP/ADP transporter, not a UDP-galactose transporter’. I do not see that as proven. Further in lines 586-588: ‘the mutation of one amino acid (F81V) of PfANT in P. falciparum resulted in resistance to two antimalarial imidazolopiperazines, KAF156 and GNF179’ Again, I do not see the evidence upon which this conclusion is based.

Table 1: The column heading ‘Inhibit (%)’ should be’ uptake (% of control)’.

Table 1 legend: ‘a mixture of 25 nM [α-32P] ATP and One-hundred-fold potential substrates (Non-radioactive) was used’ would read better as ‘a mixture of 25 nM [α-32P] ATP and non-radioactive potential substrate at one-hundred-fold concentration was used’

Figs 3B,C: The outflow of ATP and ADP is partial, rapidly plateauing. Is the rest intracellular 32P-AMP (and thus trapped inside the cells) or is there a different explanation?

Line 234-236: ‘Because TgANT has multiple transmembrane domains and both the N-terminal and C-terminal face the ER lumen, it was failed to detect Ty epitope by western blot.’ Please explain. It seems to me that the IFS might have issues with that but the Western blot should work.

I see references to Figure S1B and S1C but not S1A.

Line 517: ‘If TgANT is only the UDP-galactose transporter, then theoretically, the lack of TgANT will not cause the fatal growth defect of T.gondii.’ The reasoning is astute and likely correct - but unproven. As such a phrasing like ‘...would likely not cause a fatal growth defect in T. gondii.’ is more appropriate.

Line 539-541: ‘Since these two highly conserved domains exist in T. gondii, Plasmodium, HeLa cells, and C.elegans, we speculate that the ER ATP/ADP transport mechanism is shared among different species.’ That is overstating a bit. Even if it can be taken that these domains likely represent a signature for an ATP / ADP exchanger, the localisation is a matter of other sequences. The authors do not make the case that these domains are necessarily of ER-located transporters.

Line 575: ‘Unlike the ER ATP/ADP transporter in mammals, Apicomplexan Parasites and nematodes’ no need to capitalize Apicomplexan and Parasites.

Discussion: In my opinion the discussion about Arabidopsis, especially regarding ROS, can be shortened.

PLOS authors have the option to publish the peer review history of their article (what does this mean?). If published, this will include your full peer review and any attached files.

Reviewer #1: No

Reviewer #2: No

Reviewer #3: No
---

## [Decision Letter · Decision Letter 1]

25 May 2022

Dear Dr. Fang,

Thank you very much for submitting your manuscript "A new adenine nucleotide transporter located in the ER is essential for maintaining the growth of Toxoplasma gondii" for consideration at PLOS Pathogens. As with all papers reviewed by the journal, your manuscript was reviewed by members of the editorial board and by several independent reviewers. The reviewers appreciated the attention to an important topic. Based on the reviews, we are likely to accept this manuscript for publication, providing that you modify the manuscript according to the review recommendations.

The reviewers were satisfied with many of the changes to the manuscript, however, two additional issues remain:

1. The corrections listed by R1 should be addressed.

2. Statements firmly concluding that ER ATP levels are reduced in mutant/knockdown parasites should be modified because ER ATP levels were not directly measured. The statement at the end of the abstract is fine: "Our results indicate that TgANT is the only ATP/ADP transporter in the ER of T. gondii, and the lack of ATP in the ER is the cause of the death of T. gondii." The statement in the concluding paragraph should be modified: "The depletion of TgANT leads to the supply failure of ATP in the ER, which ..." Similarly, the section title "The interruption supply of the ER ATP caused the death of T. gondii" should be changed and any other similar statements in the manuscript.

Sincerely,

Sean T Prigge

Guest Editor

PLOS Pathogens

Dominique Soldati-Favre

Section Editor

PLOS Pathogens

Kasturi Haldar

Editor-in-Chief

PLOS Pathogens

orcid.org/0000-0001-5065-158X

Michael Malim

Editor-in-Chief

PLOS Pathogens

orcid.org/0000-0002-7699-2064

The reviewers were satisfied with many of the changes to the manuscript, however, two additional issues remain:

1. The corrections listed by R1 should be addressed.

2. Statements firmly concluding that ER ATP levels are reduced in mutant/knockdown parasites should be modified because ER ATP levels were not directly measured. The statement at the end of the abstract is fine: "Our results indicate that TgANT is the only ATP/ADP transporter in the ER of T. gondii, and the lack of ATP in the ER is the cause of the death of T. gondii." The statement in the concluding paragraph should be modified: "The depletion of TgANT leads to the supply failure of ATP in the ER, which ..." Similarly, the section title "The interruption supply of the ER ATP caused the death of T. gondii" should be changed and any other similar statements in the manuscript.

Reviewer Comments (if any, and for reference):

Reviewer's Responses to Questions

**Part I - Summary**

Reviewer #1: The manuscript has improved, with many important details that were lacking previously, as well as important data on PfANT, now included. It constitutes a significant and original contribution to the field.

Reviewer #2: The authors added a series of experiments to address the reviewers' comments and concerns. It is highly appreciated. The authors encountered a technical challenge in expressing ATP sensors in the parasite's ER. Hence, the evidence supporting the conclusion that TgANT1 transports ATP into ER is still missing, which is a key part of this manuscript. Are there any other strategies proving that the ATP content in the ER of WT is higher than that in the knockdown strain?

Reviewer #3: As I stated on the original submission, the research and the findings are original and of significant importantce to the field. I had only a series of relatively minor issues and queries, all of which have been assiduously addressed.

As such, I now recommend acceptance of this manuscript.

**Part II – Major Issues: Key Experiments Required for Acceptance**

Reviewer #1: My recommendations are relatively minor in nature and are outlined in Part III.

Reviewer #2: See the summary

Reviewer #3: none

**Part III – Minor Issues: Editorial and Data Presentation Modifications**

Reviewer #1: The localisation images have improved. However, all proteins used as markers (and their localisations) should be referred to in the relevant Figure legends (e.g. IMC and GAP45 in Fig. 1C, ALD in Fig. 4C).

There remains a need for editing, and for the following corrections/clarifications:

Line 56/57: “protein misleading” – should this be “protein misfolding”?

Line 89 – should specify what type of mammalian cells (HeLa?) are being referred to.

Line 108 – “dense granules” should not be referred to as proteins.

Line 130 – what were the CRISPR phenotype value and TMD number set to?

Line 175 – “time linear import of at least 20 min” – the uptake of ATP is not linear for this amount of time according to the graph.

Line 185 – “Kinetics of” should be replaced with “Time course for”.

A number of Figure legends (e.g. Fig. 2, Fig. 6, Fig. 9, Fig. 10) now specify that three independent experiments were performed; however, it still needs to be stated what the data shown are (mean +/- SEM of the three independent experiments?)

Line 206 – In this section, as well as in the Fig. 3A legend, it is important to state from where (the extracellular medium?), and at what time point, the presence of radioactivity is being tested for.

Do the terms “disrupted [E. coli]” and “broken [E. coli]” mean lysed?

Line 253 – area relative to what (i.e. what are the plaque sizes being compared to)?

Would the data in Fig. 5G be better placed in Fig. 4, since they relate to the growth phenotype of TgANT knockdown parasites, and not to TgANT complementation?

Given that the authors have not investigated the kinetics of TgANT or PfANT or their substrate affinities, they should not refer to the “affinity” for substrates (as they do in lines 425, 427, 604, 605). It would be more accurate to refer to the degree of inhibition caused by the various non-radiolabelled molecules at the concentration tested.

Lines 575 – 579: the information in these lines that does not come from this study should be accompanied by references.

Line 589: “the ER stress caused by ER ATP/ADP transporter” – should this be "the ER stress caused by a deficiency of the ER ATP/ADP transporter"?

Reviewer #2: None

Reviewer #3: none

PLOS authors have the option to publish the peer review history of their article (what does this mean?). If published, this will include your full peer review and any attached files.

Reviewer #1: No

Reviewer #2: No

Reviewer #3: **Yes: **Harry P. De Koning

Figure Files:

Data Requirements:

Reproducibility:

References:

---

## [Editor Report · Decision Letter 2]

4 Jun 2022

Dear Dr. Fang,

Thank you very much for submitting your manuscript "A new adenine nucleotide transporter located in the ER is essential for maintaining the growth of Toxoplasma gondii" for consideration at PLOS Pathogens. As with all papers reviewed by the journal, your manuscript was reviewed by members of the editorial board and by several independent reviewers. The reviewers appreciated the attention to an important topic. Based on the reviews, we are likely to accept this manuscript for publication, providing that you modify the manuscript according to the review recommendations.

The last sentence of the discussion still makes it sound like ATP levels were measured and found to be lacking in the ER: "The depletion of TgANT leads to the lack of ATP in the ER, which may explain why the iTgANT strain has serious growth defects."

Maybe this sentence could be modified to something like: "The depletion of TgANT likely leads to a lack of ATP in the ER, which may explain why the iTgANT strain has serious growth defects."

Sincerely,

Sean T Prigge

Guest Editor

PLOS Pathogens

Dominique Soldati-Favre

Section Editor

PLOS Pathogens

Kasturi Haldar

Editor-in-Chief

PLOS Pathogens

orcid.org/0000-0001-5065-158X

Michael Malim

Editor-in-Chief

PLOS Pathogens

orcid.org/0000-0002-7699-2064

The last sentence of the discussion still makes it sound like ATP levels were measured and found to be lacking in the ER: "The depletion of TgANT leads to the lack of ATP in the ER, which may explain why the iTgANT strain has serious growth defects."

Maybe this sentence could be modified to something like: "The depletion of TgANT likely leads to a lack of ATP in the ER, which may explain why the iTgANT strain has serious growth defects."

Reviewer Comments (if any, and for reference):

Figure Files:

Data Requirements:

Reproducibility:

References:

---

## [Editor Report · Decision Letter 3]

12 Jun 2022

Dear Dr. Fang,

We are pleased to inform you that your manuscript 'A new adenine nucleotide transporter located in the ER is essential for maintaining the growth of Toxoplasma gondii' has been provisionally accepted for publication in PLOS Pathogens.

Best regards,

Sean T Prigge

Guest Editor

PLOS Pathogens

Dominique Soldati-Favre

Section Editor

PLOS Pathogens

Kasturi Haldar

Editor-in-Chief

PLOS Pathogens

orcid.org/0000-0001-5065-158X

Michael Malim

Editor-in-Chief

PLOS Pathogens

orcid.org/0000-0002-7699-2064

The authors have now addressed the issues which arose during review and the manuscript is acceptable for publication.
---

## [Editor Report · Acceptance letter]

30 Jun 2022

Dear Dr. Fang,

We are delighted to inform you that your manuscript, "A new adenine nucleotide transporter located in the ER is essential for maintaining the growth of Toxoplasma gondii," has been formally accepted for publication in PLOS Pathogens.

Best regards,

Kasturi Haldar

Editor-in-Chief

PLOS Pathogens

orcid.org/0000-0001-5065-158X

Michael Malim

Editor-in-Chief

PLOS Pathogens

orcid.org/0000-0002-7699-2064